# Body image disturbance and associated eating disorder and body dysmorphic disorder pathology in gay and heterosexual men: A systematic analyses of cognitive, affective, behavioral und perceptual aspects

**Michaela Schmidt**[1]◉*, **Christoph O. Taube**[1]◉, **Thomas Heinrich**[1]‡, **Silja Vocks**[1]‡, **Andrea S. Hartmann**[2]◉

**1** Institute of Psychology, Department of Clinical Psychology and Psychotherapy, Osnabrück University, Osnabrück, Germany, **2** Department of Psychology, Unit of Experimental Clinical Psychology, University of Konstanz, Konstanz, Germany

◉ These authors contributed equally to this work.
‡ TH and SV also contributed equally to this work.
* micschmidt@uni-osnabrueck.de

**Data Availability Statement:** The minimal anonymized data set necessary to replicate our

## Abstract

### Objective

This study contributes to the quantitatively large, yet narrow in scope research on body image in gay men by assessing whether gay and heterosexual men systematically differ on various dimensions of body image disturbance and associated pathology, i.e., eating disorder and body dysmorphic disorder symptoms. Moreover, we examined the influence of general everyday discrimination experiences and involvement with the gay community on body image.

### Method

$N$ = 216 men ($n$ = 112 gay men, $n$ = 104 heterosexual men) participated in an online survey measuring the discrepancy between self-rated current and ideal body fat/ muscularity; drive for leanness, muscularity, and thinness; body satisfaction; body-related avoidance and checking; appearance fixing; overall body image disturbance; eating disorder and body dysmorphic disorder pathology; general everyday discrimination experiences; and involvement with the gay community.

### Results

Gay men showed a greater discrepancy between self-rated current and ideal body fat; higher drive for thinness, body-related avoidance, appearance fixing, overall body image disturbance, eating disorder and body dysmorphic disorder pathology; and lower body appreciation than heterosexual men (all $p \leq .05$). Contrary to expectation, everyday discrimination experiences were more strongly associated with body image disturbance and eating

study findings can be downloaded from a public repository (Open Science Framework). See: DOI 10.17605/OSF.IO/KFYZ7.

**Funding:** The authors received no specific funding for this work.

**Competing interests:** The authors have declared that no competing interests exist.

disorder/ body dysmorphic disorder pathology in heterosexual men than in gay men (all $p \leq$ .05). Gay community involvement was not associated with any body image disturbance-, ED-, or BDD aspect in gay men (all $p \geq$ .20).

## Discussion

The results suggest greater body image disturbance in gay men than in heterosexual men regarding cognitions, emotions, behaviors, and perception as well as higher eating disorder and body dysmorphic disorder pathology. The results also suggest the dilemma of a thin, yet muscular body ideal in gay men. Surprisingly, discrimination experiences and involvement with the gay community did not explain differences in body image disturbance. Gay men may have become resilient to discrimination over time, and body ideals might differ across gay sub-communities.

## Introduction

Body image disturbance is a complex, multidimensional construct consisting of a perceptual, a cognitive-affective, and a behavioral component [1]. The perceptual component manifests as an overestimation of one's body dimensions (e.g., body size and fat [2]) or an underestimation of one's muscularity [3], while the cognitive-affective component comprises negative thoughts, attitudes, and feelings towards one's own body, such as body dissatisfaction, shame, or disgust [4]. The behavioral component refers to body-related behaviors such as body-related avoidance or checking behavior [5] as well as investment in one's body in terms of extreme exercise behavior [6], an unhealthy obsession with healthy nutrition [7], and appearance fixing [8]. As such, body image disturbance also is a hallmark feature of eating disorders (EDs) [9] and body dysmorphic disorder (BDD) [10] and a risk factor for the development and maintenance of EDs [11].

A representative German study by Buhlmann and colleagues found that 27% of men in the study reported at least one body-related concern [12], and men are increasingly seeking psychological help for body image problems [13–15]. Despite these findings, however, previous research on body image disturbance, EDs, and BDD has mainly focused on women rather than men. It is assumed that gay men are at particular risk of developing body image disturbance and associated psychopathologies. For example, a quantitative synthesis of 30 years of research findings on body dissatisfaction and sexual orientation found significantly higher body dissatisfaction in sexual minority men than in heterosexual men [15], that might be similarly high [16] or even higher [17] than in heterosexual women. However, previous research on men's sexual orientation and body image disturbance has mostly been limited to the analysis of singular aspects (i.e., body dissatisfaction), ignoring the complexity of body image disturbance as described above.

The vast majority of research examining differences in body image disturbance between gay and heterosexual men is limited to the cognitive-affective component, mainly by assessing body dissatisfaction. These studies have yielded a consistent picture of greater body dissatisfaction in gay men than in heterosexual men [18–20]. In more detail, gay men seem to strive more strongly for a thin body (i.e., low body weight) [17, 21, 22], although some studies have reported similar levels of drive for thinness between gay and heterosexual men [20]. At the same time, there appears to be no difference between gay and heterosexual men in drive for

muscularity (i.e., a muscular, broad physique) [19, 20]. However, previous studies did not differentiate their results in terms of muscle-related cognitions and muscle-related behaviors, with the latter being better categorized as part of behavioral body image disturbance [3]. Concerning the drive for a lean body (i.e., a trained, tight physique), only one comparative study exists, indicating that gay men have a stronger drive for leanness than heterosexual men [23].

In contrast to the cognitive-affective component of body image, only a small number of studies have focused on the behavioral component of body image disturbance in gay men, and if so, predominantly on exercise behavior. In sum, results are rather divergent, with some studies suggesting that gay men exercise more often than heterosexual men [24, 25], which would be in line with a previously mentioned high drive for muscularity, and other studies revealing that gay men undertake equally [22] or even significantly less physical exercise [17, 22, 26]. Despite the multiple, yet divergent results on exercise behavior, only one study has examined differences between gay and heterosexual men regarding body-related avoidance and compulsive self-monitoring, including checking behavior [21], and found greater levels of avoidance and checking behavior in gay men. So far, there have been no comparative studies between gay and heterosexual men regarding investment in one's own body in terms of appearance fixing. Thus, those behaviors typically found for body dissatisfied women [27] have been neglected so far.

While research investigating cognitive-affective and behavioral components of body image disturbance predominantly uses self-report questionnaires, the question of how best to measure the perceptual component is contentious [4], especially with respect to self-report studies, in which there is no objective measure of body size. Such studies often employ figure rating scales [28], which conceptualize the discrepancy between one's self-rated current and ideal figure as a distortion of perception. The majority of studies found no difference between gay and heterosexual men [e.g., 29]. However, most of this research used figure rating scales in which the presented body only varies in terms of body fat and not in terms of muscularity. Considering that body discontent in men seems to be especially focused on muscularity [30], this drastically limits the findings. Only one study used a figure rating scale that represents both a body fat and a muscularity dimension, and found no group differences between heterosexual and gay men on either of the two dimensions [31].

Given the aforementioned association between body image disturbance and the development and maintenance of EDs [11] as well as BDD [10], differences in body image disturbance between gay and heterosexual men might also be mirrored by differences in ED and BDD pathology. Studies have indeed found higher prevalence rates of EDs in gay men compared to heterosexual men [32, 33] as well as a more pronounced ED pathology [e.g., 17, 34]. Also, gay men appear to exhibit more severe ED symptoms such as binge eating [34], purging behavior [35], restrictive eating [31, 36], and taking weight-reducing supplements [35]. While research on ED pathology in gay men is quite robust, research comparing gay and heterosexual men regarding BDD is limited to one study. Boroughs, Krawczyk and Thompson [37] found comparable prevalence rates for BDD in both groups, but stronger BDD pathology in gay men compared to heterosexual men. In another study, almost half of sexual minority men screened positively for BDD, a prevalence that is drastically higher than in the general population [38]. Regarding muscle dysmorphia, a subtype of BDD characterized by a pathological concern about one's muscularity, evidence is equally limited. For instance, in a validation study of the Muscle Dysmorphic Disorder Inventory (MDDDI [39]), sexual minority men reported qualitatively higher MDDI total scores than heterosexual men [28]. Furthermore, in a recent Italian study [40] nearly 9% of sexual minority men exhibited a high risk of being diagnosed with muscle dysmorphia, which was again, higher than that found in heterosexual samples.

The aforementioned differences in body image and related psychopathologies between gay and heterosexual men have been associated with several minority stress factors, such as everyday discrimination due to sexual orientation [16, 41, 42]. For example, a report published by the European Union Agency for Fundamental Rights (FRA) showed that gay men experience high levels of discrimination in everyday life [43], creating a stressful social environment that can lead to mental health problems [42]. Moreover, gay men report being exposed to discrimination due to their sexual orientation already in adolescence [44], and this early experience is suggested to be a factor in the development of greater body dissatisfaction in gay men [16]. Furthermore, perceived discrimination was reported to predict disordered eating in gay men [38], and bullying victimization was found to be associated with more coping-motivated eating [45]. Additionally, a recent study reported an indirect association between perceived stigma of gay men and proneness to an ED, mediated by self-compassion [46]. One study has also linked discrimination to higher levels of BDD symptoms or a BDD-positive screening in sexual minority men [38].

Besides discrimination, it is assumed that pressure from within the gay community to be attractive and muscular might also contribute to elevated body image concerns among gay men [47, 48]. This is in contrast to findings regarding the lesbian community which seems to act as a protective factor in the development of body dissatisfaction and appearance-related concerns [49, 50]. According to the intraminority stress theory [47], masculinity and attractiveness are means to gain status among the gay community, leading to appearance-based comparisons and competition with other community members, as well as pressure to conform to an attractive and muscular body ideal. This pressure is said to be further reinforced as men report to value attractiveness in a partner to a great extent [51] and gay and bisexual men usually rely on other men from within their sexual minority community for sexual and social relationships [47]. Therefore, gay community involvement has been linked to negative body image outcomes in gay men [48]. For example, Hospers and Jansen [52] found increased pressure to conform to appearance standards in order to attract sexual partners within the gay community. Furthermore, Convertino et al. [53] reported elevated rates of disordered body image behaviors and concerns depending on gay community involvement, and Beren et al. [16] reported an increased pressure to diet for gay men with high community involvement. Additionally, involvement with the gay community has been associated with appearance-related concerns [54], body dissatisfaction [55], and a stronger drive for muscularity [56]. However, other studies did not show an association of gay community involvement with body dissatisfaction [57], drive for muscularity [58], and drive for thinness [56], or revealed that greater levels of participation in gay-affirmative community events predicted lower body dissatisfaction [59]. That same study even found that greater alienation from the gay scene predicted increased body dissatisfaction and drive for thinness in men [59]. These findings support minority stress theories suggesting a buffering effect of gay and lesbian community involvement for sexual minority men [42]. Regarding ED pathology, Williamson and Spence [59] reported no association between participation in the gay scene and eating disturbances, whereas more frequent participation of gay men in gay-affirmative community events predicted lower eating disturbance. So far, no study has investigated affiliation with the gay community as a factor influencing the association between sexual orientation and BDD.

Overall, there is solid research suggesting an association between minority stress factors such as discrimination experiences and involvement with the gay community and body dissatisfaction and ED symptoms in gay men. Yet again, studies analyzing associations with other facets of body image disturbance as well as with BDD are scarce or non-existent, or results are highly divergent.

In sum, past research hints at stronger body image disturbance as well as higher ED and BDD pathology in gay men compared to heterosexual men. However, previous studies on this topic have mainly focused on the analyses of body dissatisfaction, as the cognitive-affective component of body image disturbance, and underlying influencing factors still remain unclear. Results regarding the behavioral or perceptual component of body image disturbance are still limited or inconsistent, and some aspects have not been investigated at all (e.g. appearance fixing). Given the multidimensional complexity of body image disturbance and the high relevance of perception distortion and behavioral coping strategies in the development and maintenance of body image disturbance as well as EDs and BDD [60], this lack of research is somewhat surprising. A comprehensive understanding of a multidimensional range of body image disturbance facets and their association with minority stress factors is essential in order to tailor integrated models of body image disturbance and adapt existing interventions for EDs and BDD for men of different sexual orientations. Therefore, in the present study, we aimed to extent the research on body image disturbance in gay men by performing a systematic multidimensional analysis of the cognitive-affective, behavioral, and perceptual component of body image disturbance and associated pathology (ED, BDD) in gay and heterosexual men. In addition, we examined everyday discrimination experiences and involvement with the gay community as potentially associated minority stress factors.

Based on the aforementioned findings, we hypothesized that gay and heterosexual men would show an equal discrepancy between their self-rated current and ideal figure in terms of body fat and muscularity (perceptual component of body image disturbance). However, we predicted that gay men would show significantly greater drive for thinness and drive for leanness, and significantly less body satisfaction compared to heterosexual men, but equally elevated cognitive drive for muscularity (cognitive-affective component of body image disturbance). We also expected significantly more body-related coping (body-related avoidance, appearance fixing) and checking behavior in gay men than in heterosexual men. However, we expected gay and heterosexual men to show equal levels of behavioral drive for muscularity (behavioral component of body image disturbance). Moreover, we hypothesized greater overall body image disturbance, ED pathology, and BDD pathology for gay men than for heterosexual men. Furthermore, we predicted that gay men, because of their sexual orientation, would report more everyday discrimination experiences than heterosexual men, and that frequency of everyday discrimination would be associated with the above-mentioned body image disturbance aspects as well as ED and BDD pathology in both groups. Finally, we predicted that in gay men, a strong involvement with the gay community would be associated with higher scores on measures of body image disturbance, ED and BDD pathology. To clarify, we defined gay community as a group of people with the shared characteristic of being gay, and gay community involvement as engaging with other members of the gay community and active participation in gay community spaces and activities, such as attending pride events, visiting a gay bar or reading a gay newspaper [55, 61].

## Materials and methods

### Participants

Data was derived from a broader online survey on body image and sexual orientation. Participants of all genders aged 18 years and older were recruited from 04/2017 to 01/2018 in German-speaking countries via university e-mail distribution lists, posters, flyers, press releases, with a particular focus on lesbian, gay, bisexual, and transgender (LGBT) websites and Facebook groups. A total of $N = 6058$ participants viewed the landing page of the online survey on unipark.de (Enterprise Feedback Suite (EFS) Survey, Questback), of whom $n = 2037$ actually

started the survey. The whole questionnaire battery was completed by $n = 838$ participants. Of the $n = 262$ men who completed the questionnaire battery, $n = 38$ men were excluded as they reported a sexual orientation other than being gay or straight (with the cell count of other sexual orientations being too low for further analysis). A further eight participants were excluded as they entered answers throughout the survey that were beyond the range from which they could choose, indicating a typing error. Further visual observation of data did not detect any conspicuous answering patterns. Thus, $n = 216$ men were included in the present analyses ($n = 112$ gay men and $n = 104$ heterosexual men).

Comparable data on body image and sexual orientation in women have already been published [51, 62]. Furthermore, the men sub-sample analyzed in the present paper was in part also included in publications focusing on the validation of the *Body Image Matrix of Thinness and Muscularity – Male Bodies* (BIMTM-MB) [63], which is not included in the present paper, and the analysis of appearance-related partner preferences and body image in men and women across sexual orientations based on the BIMTM-MB [64]. Data from the *Body Image Coping Strategies Inventory* (BICSI) [65], the *Gender-Neutral Body Checking Questionnaire* (GNBCQ) [66], the *Identification and Involvement with the Gay Community Scale* (IGCS) [61] and *The Everyday Discrimination Scale* (EDS) [67] have not yet been presented for our men sub-sample in any other study.

## Procedure

The study was reviewed and approved by Osnabrück University Ethics Committee (Ethikkommission der Universität Osnabrück). After carefully selecting the instruments that we wanted to use in our online study, we used Brislin's [68] back-translation to translate measures where no German language version was available. A bilingual translator blindly translated (i.e., forward-translated) the original English language measures, including instructions and response categories, to German. Then, a second bilingual translator independently back-translated the instrument from German to the original English language. Afterwards, the two language versions of the measurement (i.e., the original English and back-translated English versions) were compared for conceptual, item, semantic, and operational equivalence. If discrepancies occurred, another translator would try to retranslate the relevant item. This process was continued until all bilingual translators agreed that the two versions of the instrument are identical in conceptual meaning.

During the online study, participants were first informed about the study's objectives, duration, and confidentiality aspects as well as inclusion criteria. After confirming informed consent, participants started the questionnaire battery (described in alphabetical order below). The average processing time was 38 minutes. Upon completion, participants were given the opportunity to take part in a lottery to win an online shopping voucher (1 out of 10, worth 20 Euros). No further compensation for participation was payed.

## Measures

**Body Appreciation Scale-2 (BAS-2).** The BAS-2 [69; German translation available from the author] assesses general body satisfaction (e.g., appreciation, respect, and acceptance for one's own body). The scale comprises ten gender-neutral items which are rated on a 5-point Likert scale from *never* (1) to *always* (5), with higher scores indicating greater body appreciation. The internal consistency of the BAS-2 is excellent ($\alpha \geq .90$ [58]; current study: $\alpha = .92$, 95% CI [0.91, 0.94] (complete sample); $\alpha = .91$, 95% CI [0.88, 0.93] (gay sample); $\alpha = .93$, 95% CI [0.91, 0.95] (heterosexual sample)). The BAS-2 has been validated for the use with sexual minority men and women [70].

**Bodybuilder Image Grid-Original (BIG-O).**   The BIG-O [28] is a two-dimensional figure rating scale to measure perceptual body image disturbance in men. It consists of 30 drawn figures which vary along the two scales of body fat (columns) and muscularity (rows). The figures increase in body fat from left to right, and in muscularity from top to bottom, both from *extremely low* (1) to *extremely high* (5). For muscularity, larger values indicate greater muscle mass; for leanness, smaller values indicate less body fat. In the present study, participants were asked to choose the figure they think best represents their current and ideal body type. Next, the discrepancy current and ideal body fat / muscularity was calculated as a measure of perceptual body image disturbance. Larger discrepancies indicate greater perceptual body image disturbance. The test-retest reliability (one-week period) of current and ideal body types regarding body fat and muscularity is considered to be high ($.77 \leq r \leq .96$). The BIG-O is validated for the use with men [28] and has been used in studies investigating body image in gay men (e.g., [31]).

**Body Image Coping Strategies Inventory (BICSI).**   The BICSI [65; German translation available from the author] identifies how individuals deal with events and circumstances that can threaten their body image. The 29 items are rated on a 4-point scale from *definitely not like me* (0) to *definitely like me* (3), and can be allocated to the three subscales appearance fixing (10 items), avoidance (eight items), and positive rational acceptance (11 items). Larger values indicate greater coping behavior. Due to the specific research interest of the present study, only the first two subscales were used. These show a good to excellent (appearance fixing: α = .91; current study: α = .84, 95% CI [0.81, 0.87] (complete sample); α = .83, 95% CI [0.78, 0.88] (gay sample); α = .82, 95% CI [0.77, 0.87] (heterosexual sample) and acceptable (avoidance: α = .74; current study: α = .76, 95% CI [0.71, 0.81] (complete sample); α = .78, 95% CI [0.71, 0.83] (gay sample); α = .73, 95% CI [0.65, 0.80] (heterosexual sample) internal consistency. The BICSI is validated for the use in men [65] but has not yet been validated or used for a gay sample.

**Body Image Disturbance Questionnaire (BIDQ).**   The BIDQ [71] measures the extent of body image disturbance including appearance concern, preoccupation, perceived distress, functional impairment, and avoidance. It consists of 12 items, of which seven are rated on a 5-point Likert scale from *not at all concerned/ not at all preoccupied/ no distress/ no limitation/ never* (1) to *extremely concerned/ extremely preoccupied/ extreme and disabling/ extreme, incapacitating/ very often* (5). Larger values indicate greater body image disturbance. The remaining five additional qualitative open-ended items were not used in the present analyses. The BIDQ shows a good to excellent internal consistency (α = .92; current study: α = .88, 95% CI [0.85, 0.91] (complete sample); α = .87, 95% CI [0.83, 0.91] (gay sample); α = .89, 95% CI [0.85, 0.93] (heterosexual sample)). The BIDQ is validated for the use in men [72] but has not yet been validated or used for a gay sample.

**Dysmorphic Concern Questionnaire (DCQ).**   The DCQ [73] is a screening instrument for BDD. It comprises seven items that are rated on a 4-point scale from *not at all* (0) to *much more than other people* (3). Larger values indicate greater dysmorphic concerns. The DCQ shows a good internal consistency (α = .85; current study: α = .83, 95% CI [0.79, 0.86] (complete sample); α = .81, 95% CI [0.77, 0.86] (gay sample); α = .83, 95% CI [0.78, 0.88] (heterosexual sample)). The DCQ has been validated for the use with a sexual minority sample [74].

**Drive for Leanness Scale (DLS).**   The DLS [75; German translation available from the author] identifies the desire for a lean body, defined as low body fat and visible muscularity. The six items are scored on a 6-point scale from *never* (1) to *always* (6), with larger values indicating greater drive for a lean body. The questionnaire shows an acceptable to good internal consistency (α = .77; current study: α = .87, 95% CI [0.84, 0.89] (complete sample); α = .89, 95% CI [0.85, 0.92] (gay sample); α = .87, 95% CI [0.79, 0.89] (heterosexual sample)). The DLS is validated for the use in men [76] but has not yet been validated or used for a gay sample.

**Drive for Muscularity Scale (DMS).** The DMS [3] reflects the striving for a more muscular body on the two subscales *muscle-related cognitions* and *muscle-related behavior*. The 15 items are rated on a 6-point scale from *always* (1) to *never* (6), with larger values indicating greater drive for a muscular body. Item 10 "I could imagine taking anabolic steroids" was excluded from the present study due to its poor factorial validity [59]. The internal consistency of the scale is considered good to excellent (α = .90; current study: α = .89, 95% CI [0.86, 0.91] (complete sample); α = .90, 95% CI [0.86, 0.92] (gay sample); α = .88, 95% CI [0.84, 0.91] (heterosexual sample); subscale *muscle-related cognitions*: α = .90, 95% CI [0.88, 0.92] (complete sample); α = .92, 95% CI [0.89, 0.94] (gay sample); α = .88, 95% CI [0.84, 0.91] (heterosexual sample); subscale *muscle-related behavior*: α = .83, 95% CI [0.79, 0.86] (complete sample); α = .84, 95% CI [0.79, 0.88] (gay sample); α = .82, 95% CI [0.76, 0.87] (heterosexual sample)). The DMS has been validated for the use with sexual minority men [77].

**Drive for Thinness Scale (DTS).** The DTS (subscale of the Eating Disorder Inventory-2, EDI-2) [78] measures the desire to become thinner as well as the fear of gaining weight. The seven items are rated on a 6-point scale from *never* (1) to *always* (6), with larger values indicating greater drive for thinness. The DTS shows a good internal consistency (α = .85; current study: α = .88, 95% CI [0.86, 0.91]; α = .85 (complete sample); α = .89, 95% CI [0.86, 0.92] (gay sample); α = .85, 95% CI [0.81, 0.89] (heterosexual sample)). The DTS has not yet been validated for the use in a gay sample, but has been used in studies investigating body image in gay men (e.g., [20]).

**Eating Disorder Examination-Questionnaire (EDE-Q).** The EDE-Q [79] measures ED pathology within the past 28 days. A total of 22 items can be allocated to four subscales: *restraint* (five items), *eating concern* (five items), *weight concern* (five items), and *shape concern* (eight items). The remaining six items, which assess diagnostic features, were not included in the present analyses. The included items are rated on a 7-point Likert scale from *no days* / *none of the times* / *not at all* (0) to *every day* / *every time* / *markedly* (6). Larger values indicate greater ED pathology. The internal consistency is considered to be excellent for the overall questionnaire (α = .97; current study: α = .93, 95% CI [0.91, 0.94] (complete sample); α = .93, 95% CI [0.92, 0.95]) (gay sample); α = .92, 95% CI [0.90, 0.94]) (heterosexual sample)), and acceptable to excellent for the subscales (.85 ≤ α ≤ .93; current study: .76, 95% CI [0.71, 0.81] ≤ α ≤ .88, 95% CI [0.86, 0.91] (complete sample); .77, 95% CI [0.70, 0.83] ≤ α ≤ .88, 95% CI [0.84, 0.91] (gay sample); .75, 95% CI [0.66, 0.81] ≤ α ≤ .88, 95% CI [0.85, 0.91] (heterosexual sample)). The EDE-Q has been validated for the use with sexual minority men [80].

**Gender-Neutral Body Checking Questionnaire (GNBCQ).** The GNBCQ [66; German translation available from the author] measures gender-neutral body-checking behavior, i.e., without checking behaviors that could be conceptualized as more specific to the body image of men or women. It encompasses 10 items, which are rated on a 5-point Likert scale from *never* (1) to *very often* (5), with larger values indicating greater body-checking behavior. The internal consistency for the subgroup of men is considered to be good to excellent (α = .96; current study: α = .84, 95% CI [0.80, 0.87] (complete sample); α = .84, 95% CI [0.80, 0.88] (gay sample); α = .84, 95% CI [0.78, 0.88] (heterosexual sample)). The GNBCQ has been validated on men, but not on a sexual minority sample [66].

**Identification and Involvement with the Gay Community Scale (IGCS).** The IGCS [61; German translation available from the author] assesses gay and bisexual men's affiliation with and perceived closeness to the gay male community, such as through reading gay newspapers or attending gay-affirmative events. It consists of eight items, of which the first seven are rated on a 5-point Likert scale from *strongly disagree* (1) to *strongly agree* (5) or from *not at all* (1) to *several times a week or daily* (5). The eighth item is rated on a 5-point scale from *no gay friends* (1) to *five or more gay friends* (5). Larger values indicate greater identification and

involvement. The internal consistency of the scale is acceptable ($\alpha$ = .78; current study: $\alpha$ = .74, 95% CI [0.64, 0.80] (gay sample)).

**The Everyday Discrimination Scale (EDS).** The EDS [67; German translation available from the author] measures the frequency of universal everyday discrimination experiences, e.g., being insulted or treated differently on a regular basis. It consists of 10 items, which are rated on a 6-point Likert scale from *never* (1) to *almost every day* (6). Larger values indicate more frequent everyday discrimination experiences. The tenth item asks about the specific self-suspected reason for discrimination, like age, nationality, or sexual orientation. Therefore, the EDS does not only apply to discrimination experiences based on sexual orientation. The scale shows a good internal consistency ($\alpha$ = .88; current study: $\alpha$ = .87, 95% CI [0.84, 0.89] (complete sample); $\alpha$ = .84, 95% CI [0.80, 0.88] (gay sample); $\alpha$ = .89, 95% CI [0.86, 0.92] (heterosexual sample)). The EDS has not yet been validated or used on a gay sample.

**Sociodemographic characteristics.** Sexual orientation was measured via self-report. Participants were able to choose from a range of different categories of sexual orientations (gay, lesbian, heterosexual, bisexual, pansexual, polysexual, asexual), although we explicitly acknowledged that sexual orientation is a continuum. If none of the categories met their sexual orientation, participants could type in their sexual orientation in a text field. Further data were gathered on age, gender, nationality, relationship status, highest educational attainment, body height (in meters) and weight (in kilograms) in order to calculate body mass index (BMI), again via self-report.

## Statistical analyses

All analyses were performed using SPSS Statistics (Version 26) [81] except for the two one-sided *t*-tests to test for equivalence between groups, which were run with the open source software *jamovi* (Version 1.6) [82]. To compare groups in terms of demographic characteristics, we used $\chi^2$ tests (or Monte Carlo exact tests with 10,000 samples and 99% confidence interval if more than 20% of expected frequencies were between 1 and 5) or *t*-tests for independent groups. In case of a significant $\chi^2$ test, adjusted residuals were calculated and checked to locate the source of the significance. An adjusted residual with an absolute value that exceeded +/- 1.96 indicated lack of fit of the null hypothesis, i.e., significance [83].

To test the expected group differences in body image disturbance facets, ED and BDD pathology, and the frequency of discrimination experiences, we again conducted *t*-tests for independent groups. In the case of heterogeneity of variance, Welch's tests were employed. We adjusted the *p*-values with Benjamini-Hochberg correction to correct for multiple testing [84]. Effect sizes were reported as Cohen's *d* (small effect: *d* = 0.2, medium effect: *d* = 0.5, large effect *d* = 0.8; [85]; for *t*-tests) or Cramér's *V* (small effect: *V* = 0.1, medium effect: *V* = 0.3, large effect *V* = 0.5; [85]; for $\chi^2$ tests). To test for equivalence of groups, two one-sided *t*-tests (TOST) were calculated. The test is a variation of the standard one-sided t-test, that examines whether the hypothesis that the difference between two groups is zero can be rejected. The TOST, however, examine whether the hypothesis that the difference between groups is meaningful (i.e., at least as extreme as the smallest effect size of interest) can be rejected. The smallest effect size of interest was set using established benchmarks [86], namely at *d* = 0.2, which represents a trivially small effect size [85]. Groups are considered equivalent when both of the two one-sided *t*-tests are statistically significant. In case the TOST was non-significant, indicating that groups are not statistically equivalent, a one-sided t-test was run to check if groups significantly differed from each other. In the case of heterogeneity of variance, Welch's tests were employed.

To investigate the association of frequency of discrimination experiences as well as involvement with the gay community with body image disturbance facets, ED pathology, and BDD

pathology, we calculated Spearman's correlation coefficient $\rho$ for gay men and heterosexual men separately, or in the case of involvement with the gay community for gay men only (small effect: $\rho = 0.1$; medium effect: $\rho = 0.3$, large effect: $\rho = 0.5$; [85]). Extreme outliers (more than 3 times the interquartile range) were checked for unrealistic answers or response patterns and kept in the sample if not applicable. However, we checked whether significantly divergent results emerged after eliminating these outliers and reported this if applicable.

## Results

**Sociodemographic characteristics.** Table 1 shows sociodemographic characteristics of gay men and heterosexual men. The two groups did not differ significantly in age, BMI, relationship status or nationality, but did differ in terms of highest educational attainment (Table 1). The observation of the adjusted residuals suggested that the rejection of the null hypothesis resulted as, compared to heterosexual men, a larger number of gay men had no higher-track secondary school qualification than statistically expected.

## Group differences in body image disturbance facets, eating disorder and body dysmorphic disorder pathology, and discrimination experiences

With regard to body image disturbance facets, gay men showed significantly higher scores in terms of drive for thinness (DTS), appearance fixing (BICSI-*appearance fixing*) and general body image disturbance (BIDQ) compared to heterosexual men, while heterosexual men scored significantly higher than gay men regarding body appreciation (BAS-2). There were no significant differences between the two groups in terms of drive for leanness (DLS), body avoidance (BICSI-*avoidance*) and body checking (GNBCQ) (Table 2). The equivalence tests

**Table 1. Group comparisons regarding demographic characteristics.**

| Variables | Gay men (n = 112) | | Heterosexual men (n = 104) | | Group Comparisons | | | |
|---|---|---|---|---|---|---|---|---|
| | *M* | *SD* | *M* | *SD* | *T* | *df* | *p* | *Cohen's d* |
| Age (years) | 30.26 | 11.31 | 28.82 | 9.76 | .95 | 214.17 | .316 | 0.14 |
| BMI (kg/m$^2$) | 24.80 | 5.26 | 24.36 | 4.21 | .63 | 211.06 | .497 | 0.09 |
| | *n* | *%* | *n* | *%* | $\chi^2$ | *df* | *p* | *Cramer's V* |
| Education | | | | | 6.91 | 2 | **.032** | .18 |
| University degree/ Polytechnic degree | 46 | 41.1 | 49 | 47.1 | | | | |
| Higher-track secondary school qualifications | 45 | 40.2 | 48 | 46.2 | | | | |
| No higher-track secondary school qualifications | 21 | 18.8 | 7 | 6.7 | | | | |
| Relationship status | | | | | 3.32 | 2 | .180 | .12 |
| In a relationship[a] | 47 | 42.0 | 50 | 48.1 | | | | |
| Not in a relationship[b] | 64 | 57.1 | 50 | 48.1 | | | | |
| Another unlisted relationship status | 1 | 0.9 | 4 | 3.8 | | | | |
| Nationality | | | | | 0.11 | 1 | .743 | .02 |
| German | 102 | 91.1 | 96 | 92.3 | | | | |
| Other | 10 | 8.9 | 8 | 7.7 | | | | |

*Note.* BMI = body mass index

*M* = mean; *SD* = standard deviation.

[a] includes committed relationship, living separately / living together; married; partnered

[b] includes single; separated; divorced; widowed

**Table 2. Group comparisons regarding body image disturbance facets, eating disorder and body dysmorphic disorder pathology, and everyday discrimination experiences.**

| Variables | Gay men (*n* = 112) | | Heterosexual men (*n* = 104) | | Group Comparisons | | | |
|---|---|---|---|---|---|---|---|---|
| | *M* | *SD* | *M* | *SD* | *T* | *df* | *p*[1] | Cohen's *d* |
| Cognitive-affective body image disturbance | | | | | | | | |
| BAS | 3.31 | 0.72 | 3.63 | 0.80 | -3.01 | 214 | **.008** | -0.42 |
| DLS | 3.85 | 1.13 | 3.78 | 1.04 | 0.48 | 214 | .675 | 0.06 |
| DTS | 2.64 | 1.14 | 2.20 | 0.88 | 3.24 | 207.25 | **.005** | 0.43 |
| Behavioral body image disturbance | | | | | | | | |
| BICSI–appearance fixing | 1.37 | 0.58 | 1.07 | 0.52 | 3.99 | 214 | **< .001** | 0.55 |
| BISCI–avoidance | 0.83 | 0.57 | 0.66 | 0.50 | 2.23 | 214 | **.045** | 0.32 |
| GNBCQ | 2.06 | 0.70 | 1.98 | 0.64 | 0.87 | 214 | .488 | 0.12 |
| Overall body image disturbance | | | | | | | | |
| BIDQ | 1.95 | 0.71 | 1.62 | 0.70 | 3.38 | 214 | **.005** | 0.47 |
| EDE-Q total score | 1.48 | 1.15 | 1.12 | 0.96 | 2.52 | 214 | **.033** | 0.34 |
| Eating disorder pathology | | | | | | | | |
| EDE-Q–restraint | 1.22 | 1.35 | 1.03 | 1.16 | 1.15 | 214 | .341 | 0.15 |
| EDE-Q–eating concern | 0.65 | 1.07 | 0.45 | 0.74 | 1.60 | 198.08 | .165 | 0.22 |
| EDE-Q–weight concern | 1.49 | 1.32 | 1.11 | 1.13 | 2.26 | 214 | **.045** | 0.31 |
| EDE-Q–shape concern | 2.03 | 1.38 | 1.49 | 1.25 | 3.02 | 214 | **.009** | 0.41 |
| Body dysmorphic pathology | | | | | | | | |
| DCQ | 6.60 | 3.90 | 5.39 | 3.97 | 2.26 | 214 | **.045** | 0.31 |
| Everyday discrimination experiences | | | | | | | | |
| EDS | 1.82 | 0.68 | 1.80 | 0.80 | 0.249 | 214 | .800 | 0.03 |

*Note. M* = mean; *SD* = standard deviation; BAS = Body Appreciation Scale-2; BICSI = Body Image Coping Strategies Inventory; BIDQ = Body Image Disturbance Questionnaire; DCQ = Dysmorphic Concern Questionnaire; DLS = Drive for Leanness Scale; DTS = Drive for Thinness Scale; EDE-Q = Eating Disorder Examination-Questionnaire; GNBCQ = Gender-Neutral Body Checking Questionnaire; EDS = The Everyday Discrimination Scale. Significant group comparisons are in bold.
[1]Benjamini-Hochberg adjusted p-values.

(TOSTs) as well as the null hypothesis tests (one-sided t-tests) regarding the discrepancy between current and ideal muscularity (BIG-O subscale) and muscle-related behavior (DMS subscale) were both non-significant. This indicates that groups were neither statistically equal, nor significantly different from each other. Hence, the difference between the two groups was somewhere between zero and the smallest effect size of interest that was previously set. As such, it is not possible to sufficiently interpret those results. The equivalence test regarding muscle-related cognitions (DMS subscale) was significant, whereas the null hypothesis test was non-significant, meaning that the observed effect was statistically equivalent to zero. For the discrepancy between current and ideal body fat (BIG-O subscale), the equivalence test was non-significant, but the null hypothesis test reached statistical significance, indicating that the observed effect was statistically different from zero (Table 3).

Concerning ED and BDD pathology, gay men scored significantly higher than heterosexual men in terms of total eating disorder pathology (EDE-Q total score) and the subscales *weight concern* and *shape concern* as well as dysmorphic concerns (DCQ). There were no group differences regarding the subscales *restraint eating* and *eating concern* (EDE-Q) (Table 2).

There was no difference between gay men and heterosexual men with regard to the frequency of everyday discrimination experiences (EDS) (Table 2). For a detailed description of self-rated suspected reasons for discrimination, see Table 4.

**Table 3. Equivalence testing regarding perceptive body image disturbance and drive for muscularity.**

| Variables | Gay men (n = 112) | | Heterosexual men (n = 104) | | Test | | | | |
|---|---|---|---|---|---|---|---|---|---|
| | M | SD | M | SD | | | T | df | p |
| Perceptual body image disturbance | | | | | | | | | |
| BIG-O–Discrepancy current—ideal muscularity | 0.66 | 0.98 | 0.71 | 0.78 | t-test | | -0.42 | 209 | .674 |
| | | | | | TOST Upper | | -1.90 | 209 | **.030** |
| | | | | | TOST Lower | | 1.05 | 209 | .147 |
| BIG-O–Discrepancy current—ideal body fat | -1.08 | 1.09 | -0.68 | 1.05 | t-test | | -2.73 | 214 | **.007** |
| | | | | | TOST Upper | | -4.20 | 214 | **< .001** |
| | | | | | TOST Lower | | -1.26 | 214 | .896 |
| Behavioral body image disturbance | | | | | | | | | |
| DMS–behavior | 1.91 | 0.95 | 2.04 | 0.92 | t-test | | -0.98 | 214 | .330 |
| | | | | | TOST Upper | | -2.45 | 214 | **.008** |
| | | | | | TOST Lower | | 0.49 | 214 | .311 |
| DMS–cognitions | 3.46 | 1.30 | 3.34 | 1.14 | t-test | | 0.77 | 213 | .440 |
| | | | | | TOST Upper | | -2.91 | 213 | .002 |
| | | | | | TOST Lower | | 4.46 | 213 | **< .001** |

*Note*. TOST = two one-sided t-tests; *M* = mean; *SD* = standard deviation; BIG-O = Bodybuilder Image Grid-Original; DMS = Drive for Muscularity Scale. Significant group comparisons are in bold.

## Correlations of everyday discrimination experiences with body image disturbance facets, eating disorder pathology, and body dysmorphic disorder pathology in gay men and heterosexual men

In terms of body image disturbance facets, everyday discrimination (EDS) of gay men was only positively correlated with body avoidance (BICSI-*avoidance*) and general body image disturbance (BIDQ). In heterosexual men, there were positive correlations of everyday discrimination (EDS) with muscle-related cognitions (DMS subscale), *appearance fixing*, body *avoidance* (both subscales of the BICSI), body checking (GNBCQ), and general body image disturbance (BIDQ). There was also a negative correlation between everyday discrimination (EDS) and body appreciation (BAS-2) in heterosexual men (see Table 5).

Concerning ED and BDD pathology, everyday discrimination (EDS) of gay men was positively correlated with total eating disorder pathology (EDE-Q total score) and the subscales

**Table 4. Self-rated suspected reason for discrimination as stated in the everyday discrimination experience scale.**

| Suspected reason | Gay men (n = 112) | | Heterosexual men (n = 104) | |
|---|---|---|---|---|
| | Total | % | Total | % |
| Sexual orientation | 63 | 55.8 | 5 | 4.8 |
| Origin/ nationality | 11 | 9.7 | 24 | 23.1 |
| Sex | 10 | 8.8 | 13 | 12.5 |
| Ethnicity | 5 | 4.4 | 12 | 11.5 |
| Age | 21 | 18.6 | 21 | 20.2 |
| Religion | 0 | 0 | 9 | 8.7 |
| Height | 12 | 10.6 | 18 | 17.3 |
| Weight | 22 | 19.5 | 17 | 16.3 |
| Other aspect of physical appearance | 17 | 15.0 | 23 | 22.1 |
| Other reason | 24 | 21.2 | 33 | 31.7 |

**Table 5. Correlations of everyday discrimination experiences with body image disturbance facets, eating disorder pathology and body dysmorphic disorder pathology.**

| Variable | Everyday discrimination experience (EDS) | | | |
| --- | --- | --- | --- | --- |
| | Gay men (n = 112) | | Heterosexual men (n = 104) | |
| | ρ | p | ρ | p |
| Perceptual body image disturbance | | | | |
| BIG-O-Discrepancy current–ideal body fat | .098 | .306 | .019 | .846 |
| BIG-O-Discrepancy current–ideal muscularity | .110 | .246 | .049 | .623 |
| Cognitive-affective body image disturbance | | | | |
| BAS | -.075 | .430 | -.263 | **.007** |
| DLS | .032 | .734 | .035 | .721 |
| DMS–cognitions | .104 | .275 | .315 | **.001** |
| DTS | .022 | .822 | .172 | .081 |
| Behavioral body image disturbance | | | | |
| BICSI–appearance fixing | .159 | .094 | .290 | **.003** |
| BISCI–avoidance | .193 | **.041** | .360 | **.000** |
| GNBCQ | .165 | .081 | .308 | **.001** |
| DMS–behavior | -.089 | .348 | .097 | .327 |
| Overall body image disturbance | | | | |
| BIDQ | .280 | **.003** | .322 | **.001** |
| Eating disorder pathology | | | | |
| EDE-Q total score | .184 | **.052** | .450 | **.000** |
| EDE-Q–restraint | .098 | .302 | .271 | **.005** |
| EDE-Q–eating concern | .121 | .202 | .422 | **.000** |
| EDE-Q–weight concern | .224 | **.018** | .452 | **.000** |
| EDE-Q–shape concern | .182 | **.055** | .405 | **.000** |
| Body dysmorphic disorder pathology | | | | |
| DCQ | .207 | **.028** | .243 | **.013** |

*Note*. BIG-O = Bodybuilder Image Grid-Original; BICSI = Body Image Coping Strategies Inventory; BIDQ = Body Image Disturbance Questionnaire; DLS = Drive for Leanness Scale; DMS = Drive for Muscularity Scale; DTS = Drive for Thinness Scale; EDE-Q = Eating Disorder Examination-Questionnaire; GNBCQ = Gender-Neutral Body Checking Questionnaire; EDS = The Everyday Discrimination Scale. Significant effects are in bold.

*weight concern* and *shape concern* as well as dysmorphic concerns (DCQ). In heterosexual men, everyday discrimination (EDS) was positively correlated with total eating disorder pathology (EDE-Q total score) and all subscales (eating concern, weight concern, shape concern, restraint eating), as well as dysmorphic concerns (DCQ) (see Table 5).

## Correlations of involvement with the gay community with body image disturbance facets, eating disorder pathology, and body dysmorphic disorder pathology in gay men

Involvement with the gay community (IGCS, *M* = 2.91; *SD* = 0.72) was not significantly associated with any of the body image disturbance facets or with ED and BDD pathology (see Table 6).

## Discussion

The objective of the present study was to extent the literature on body image in gay men by providing a multidimensional analysis of perceptual, cognitive-affective and behavioral body

**Table 6. Correlations of Involvement with the gay community with body image disturbance facets, eating disorder pathology and body dysmorphic disorder pathology.**

| Variable | Involvement with the gay community scale (IGCS) | |
|---|---|---|
| | **Gay men (n = 112)** | |
| | $\rho$ | $p$ |
| Perceptual body image disturbance | | |
| BIG-O–Discrepancy current—ideal body fat | -.007 | .938 |
| BIG-O–Discrepancy current—ideal muscularity | -.073 | .444 |
| Cognitive-affective body image disturbance | | |
| BAS | .005 | .960 |
| DLS | .070 | .464 |
| DMS–cognitions | -.112 | .242 |
| DTS | .045 | .637 |
| Behavioral body image disturbance | | |
| BICSI–appearance fixing | .001 | .989 |
| BISCI–avoidance | -.033 | .726 |
| GNBCQ | .070 | .466 |
| DMS–behavior | .053 | .578 |
| Overall body image disturbance | | |
| BIDQ | .005 | .957 |
| Eating disorder pathology | | |
| EDE-Q total score | .079 | .409 |
| EDE-Q–restraint | .057 | .554 |
| EDE-Q–eating concern | .004 | .969 |
| EDE-Q–weight concern | .040 | .679 |
| EDE-Q–shape concern | .084 | .378 |
| Body dysmorphic disorder pathology | | |
| DCQ | .006 | .954 |

*Note*. BIG-O = Bodybuilder Image Grid-Original; BICSI = Body Image Coping Strategies Inventory; BIDQ = Body Image Disturbance Questionnaire; DLS = Drive for Leanness Scale; DMS = Drive for Muscularity Scale; DTS = Drive for Thinness Scale; EDE-Q = Eating Disorder Examination-Questionnaire; GNBCQ = Gender-Neutral Body Checking Questionnaire; EDS = The Everyday Discrimination Scale.

image disturbance facets and associated ED and BDD pathology in gay and heterosexual men. Moreover, we sought to examine the association of sexual minority stress factors like discrimination experiences and involvement with the gay community with body image disturbance in gay men.

In line with our hypothesis, on the cognitive-affective dimension of body image disturbance, gay men showed significantly lower body appreciation and significantly higher drive to be thin and to lose weight compared to heterosexual men. This corroborates the solid foundation of previous research which reported higher body dissatisfaction in gay men than in heterosexual men, especially with regard to body weight [17, 19, 20]. However, there was no significant difference between gay and heterosexual men in terms of drive for leanness, i.e., a trained, tight physique with low body fat and immediately visible muscularity. Moreover, the two groups showed equal levels of cognitive drive for muscularity. The results support previous research that revealed no difference between gay and heterosexual men regarding the desire to have the "perfect" muscular body [87], and an overall trend of increased muscularity-focused body dissatisfaction in men [30, 88, 89]. At the same time, gay men might also be oriented

towards a not only lean, but thin body ideal, usually ascribed to heterosexual women [90]. It is argued that, like heterosexual women, gay men may view their bodies as sex objects to attract men, making them anxious to look not only strong, but also youthfully thin [91]. The results contradict findings proposing that the drive for a thin body is a "female" body image issue [87, 90, 92] and underlines the paradox of a highly light-weight, yet wide and muscular body ideal for gay men [17]. This rather unattainable body ideal might cause a dilemma influencing the highly elevated body dissatisfaction in gay men.

The findings described above might also be reflected by greater discrepancies between self-rated current and ideal body fat in gay men, however comparable, though not statistically equal, discrepancies between self-rated current and ideal muscularity in gay and heterosexual men found in our study. These findings correspond to earlier studies reporting greater discrepancies between self-rated current and ideal body fat in gay men than in heterosexual men [93, 94], but no difference between gay and heterosexual men regarding the discrepancies between self-rated current and ideal muscularity [31]. However, as we established neither statistical equality nor statistical differences between gay and heterosexual men regarding the muscularity dimension, our findings should be treated with caution.

With regard to the behavioral component of body image disturbance, gay men reported significantly more body-related coping strategies such as appearance-fixing or avoidance behavior, which is in line with our initial hypothesis and the so far only previous study on those aspects by Cella and colleagues [21]. As gay men consider appearance more essential to their sense of self than do heterosexual men [52], and appearance-fixing and avoidance are strategies to cope with potential threats or challenges to body image [65], it is likely that they engage more frequently in these coping strategies to protect their self-worth. Regarding behaviors that target muscularity, namely behavioral drive for muscularity, once again, gay and heterosexual men showed similar, though not statistically equal, results. This supports our hypothesis and previous research examining overall drive for muscularity [19, 20], as well as a previous study that found no difference between gay and heterosexual men in extreme exercise behavior [22]. Nevertheless, the two groups did not differ significantly in terms of checking behavior, which contradicts our predictions as well as the only previous study that has assessed checking behavior in gay and heterosexual men [21], which used the Body Uneasiness Test (BUT; [95]). The similar scores between gay and heterosexual men in our study might stem from the fact that half of the items (5/10) in our instrument (GNBCQ) explicitly refer to muscle-related checking, thus mainly pertaining to body image aspects in which the two groups do not seem to differ. The BUT, by contrast, operationalizes checking behavior more broadly (i.e., time spent in front of the mirror; difficulties to avert gaze from own body), which could account for the differential findings.

In accordance with the finding of greater body image disturbance in our study as described above, gay man also showed significantly higher ED and BDD pathology. In more detail, gay men showed higher overall ED pathology than heterosexual men, confirming our initial hypothesis and the majority of previous research, which also reported more elaborated ED pathology in gay men [20, 31, 94, 96–98]. Moreover, gay men showed higher weight and shape concern, but did not show higher restraint eating or eating concern. This indicates that although gay men seem to have more ED-related concerns about how they look and how much they weigh, they apparently do not differ from heterosexual men in terms of pathological ED-related behaviors. This contradicts our expectations and previous research indicating more dieting behavior [36], fasting [99], and greater use of diet pills in gay men [25, 35, 99, 100]. A possible explanation for these discrepant findings may be that the participants' age was much lower in previous studies (e.g., mean age of sample in years: 29,54 (our study) vs. 22.4 [35], 23.5 [36], 16.04 [99], 15.9 [100]), and eating disorder symptom severity seems to be

highest in adolescence and young adults, before declining in adulthood [101]. Regarding BDD pathology, scores were higher in gay men as well, which is in line with our initial hypothesis and the small amount of previous research on differences in BDD between gay and heterosexual men [37].

To account for expected group differences in body image disturbance facets, ED and BDD pathology, we suggested minority stress factors, such as everyday discrimination. However, gay and heterosexual men did not differ in the frequency of everyday discrimination experiences. Furthermore, for gay men, discrimination was only positively associated with the severity of BDD pathology and some ED subscales as well as overall body image disturbance, but rarely with any specific components of body image disturbance. For heterosexual men, on the other hand, we found associations between discrimination and BDD pathology, total ED pathology and all specific subscales, overall body image disturbance, body satisfaction and all aspects on the behavioral dimension of body image disturbance (i.e., behavioral drive for muscularity, body-related avoidance, checking, appearance-fixing). This indicates that everyday discrimination does not seem to have influenced the more pronounced multidimensional body image disturbance, ED and BDD pathology in gay men in this study. However, everyday discrimination does seem to affect body image and associated pathologies in heterosexual men. A possible explanation could be that the instrument we used was initially designed to measure discrimination among people of different races and ethnicities, and items mostly refer to discrimination that is based on racial stereotypes and not on stereotypes regarding sexual orientation. Thus, the instrument might not reflect common discrimination experiences of gay men. Accordingly, when gay men reported discrimination experiences, only half of them listed their sexual orientation as the perceived reason for their discrimination experiences, while for heterosexual men, the most frequent perceived reason for everyday discrimination was nationality. Furthermore, we assessed current discrimination experiences in the everyday life of gay men. However, gay men appear to suffer from stereotypes and bullying due to their sexual orientation from adolescence onwards [44], which poses a critical phase for the development and manifestation of body image disturbance [102]. It is possible that discrimination in this psychologically vulnerable life phase has an even bigger impact on body image and associated pathologies than current discrimination. Alternatively, gay men may have become resilient to discrimination over time, lessening the impact of current discrimination experiences on their body image and associated symptoms. Lastly, due to previous experiences, gay men might have come to expect stigma and anticipate discrimination due to previous experiences [42], and this anticipation may account for mental health-related distress for gay men [103].

Surprisingly, and contrary to our hypothesis, body image disturbance facets, BDD and ED pathology in gay men were not associated with the extent of involvement with the gay community, even though our sample of gay men was rather engaged with the gay community. This confirms the findings of some previous studies which reported no association of gay community involvement with body image disturbance facets such as body dissatisfaction [57]. However, it contradicts other studies which did report such associations with body dissatisfaction [55] and with drive for muscularity [56]. These contradictory findings may be explained by the different instruments that were used to measure involvement with the gay community. For instance, the IGCS used in our study not only measures participation and involvement with the gay community (e.g., attending gay-affirmative events, reading gay magazines), but also self-identification as gay and identification with the gay community. The self-constructed instrument used by Davids et al. (Gay Community Participation Scale) [55] measures frequency of involvement and participation with the gay community only. As predominantly involvement with the gay community is suspected to convey specific unrealistic body ideals that contribute to body discontent [104], this could be a possible reason for the non-significant

association of the IGCS and body image disturbance in our gay sample. Since the authors of the IGCS do not clearly define which items refer to involvement and which items refer to identification with the gay community, we were not able to calculate subscales to test for this hypothesis. Furthermore, and consistent with prior studies focusing on body image in gay men, the present study conceptualized the gay community as a broad social system. However, the body ideals of men's gay culture are rather divergent (e.g., muscular "bears" vs. youthful "twinks" [105]) and may also differ between countries and associated cultural backgrounds, contributing to the divergent findings across nationwide studies. Moreover, gay subgroups seem to differ in the extent to which members are reduced to their appearance [58], a factor which appears to mediate the association between involvement with the gay community and body dissatisfaction [55]. To assess these factors, future studies should examine body image among particular groups within the gay community. Lastly, it was argued that intracommunity pressure to conform to a certain body ideal stems from the wish to attract other members of the community for sexual and social relationships [47]. However, 42% of gay men in our study were in a committed relationship, which was a similar quantity as in heterosexual men. This may have lessened the pressure from the gay community on our gay sample to stay attractive since men in committed relationships may be less concerned with attracting new partners.

Some limitations of the present study should be mentioned. Due to the cross-sectional study design, it was not possible to investigate causal relationships between sexual orientation and our hypothesized influencing factors. Regarding the use of figural drawing scales to measure perceptual body image distortion, general limitations of these measures include that scales only display hand-drawn und therefore less detailed body images, that are based on an artist's subjective belief of varying bodies weight and muscularity. Also, those measures only depict a limited set of varying bodies, while in reality body shape is a continuous variable. Those aspects might have limited the validity of our results [63]. Furthermore, as we only used self-reports, we did not have an objective measure of participants' bodies (e.g., height, weight, muscularity, body fat), which limits the ability to draw conclusions regarding perceptual body image disturbance. Accordingly, the discrepancy between self-rated current and ideal body could also be interpreted as perceptual body discontent [106, 107] and therefore allocated to the cognitive-affective facet of body image disturbance. Also, as there were no previously validated German language versions of the Body Appreciation Scale-2 [68], the Body Image Coping Strategies Inventory [65], the Drive for Leanness Scale [75], the Gender-Neutral Body checking Questionnaire [66], the Identification with Gay Community Scale [61] and The Everyday Discrimination Scale [67], we had to translate those by ourselves via back-translation [68]. We did not conduct a comprehensive validation process of the translated measures, but the but the internal consistencies of the translated measures are similar to the original validation studies. Moreover, as the present sample is community-based, adult, non-clinical and mostly with an academic background, the results cannot be transferred to clinical and non-academic populations or adolescents. Lastly, we found neither statistical equivalence nor significant differences between the groups regarding behavioral drive for muscularity and discrepancy between self-rated current and ideal body fat. This might be due to our method of determining the smallest effect size of interest, which was based on established benchmarks (Cohen's effect size conventions [85]), and not on related studies in the literature or individual, empirical considerations [86].

The present study contributes to the quantitatively large, yet narrow in scope research on body image in gay men by systematically examining multiple dimensions of body image disturbance in gay and heterosexual men as well as associated pathologies, including the under-investigated BDD pathology. Overall, the results suggest that gay men not only show more body dissatisfaction than heterosexual men, but a significantly higher multidimensional body

image disturbance affecting cognitions, emotions, behaviors, and perception. This might be especially true for facets linked to body weight and thinness, suggesting the dilemma of a paradoxically thin, yet muscular body ideal for gay men, that might not even be dissolved in a lean body. In accordance with that, gay man also showed significantly higher ED and BDD pathology for which body image disturbance poses an eminently relevant risk factor [10, 11]. However, differences in body image might not be associated with the frequency of everyday discrimination experiences and cross-subgroup involvement with the gay community.

Our findings might be used to tailor existing models of body image and to adapt the prevention, counseling, and treatment of body image disturbance, BDD or EDs for men with different sexual orientations. For instance, counselors and therapists treating gay men should pay attention to conflicting body ideals, including men's muscle-related body ideals, but also ideals and coping strategies regarding body weight and general physical attractiveness.

## Acknowledgments

The authors thank Sarah Mannion for proof-reading the manuscript.

## Author Contributions

**Conceptualization:** Christoph O. Taube, Silja Vocks, Andrea S. Hartmann.

**Data curation:** Christoph O. Taube, Thomas Heinrich.

**Formal analysis:** Michaela Schmidt.

**Supervision:** Andrea S. Hartmann.

**Writing – original draft:** Michaela Schmidt.

**Writing – review & editing:** Christoph O. Taube, Thomas Heinrich, Silja Vocks, Andrea S. Hartmann.

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
