## [Decision Letter · Decision Letter 0]

20 Apr 2022

PONE-D-21-24159Body image disturbance, eating disorder and body dysmorphic disorder pathology in homosexual and heterosexual men: Do discrimination experiences and involvement with the gay community matter?PLOS ONE

Dear Dr. Schmidt,

Thank you for submitting your manuscript to PLOS ONE. After careful consideration, we feel that it has merit but does not fully meet PLOS ONE’s publication criteria as it currently stands. Therefore, we invite you to submit a revised version of the manuscript that addresses the points raised during the review process.

The manuscript has been evaluated by three reviewers, and their comments are available below.

The reviewers have raised a number of major concerns that need attention. In particular, please pay close attention to the requests to amend the language usage throughout to adhere to bias-free standards. The reviewers also request that the cited literature is updated to include more recent sources, and that some of the studies findings are elaborated on in the discussion.

Could you please revise the manuscript to carefully address the concerns raised?

We look forward to receiving your revised manuscript.

Kind regards,

Jamie Royle

Staff Editor

PLOS ONE

Journal Requirements:

2. We noted in your submission details that a portion of your manuscript may have been presented or published elsewhere. "Henn, Taube, Vocks, & Hartmann, 2019", "BIMTM-MB; Arkenau, Vocks, Taube, Waldorf, & Hartmann, 2020", "Cordes, Vocks, & Hartmann, in press" Please clarify whether this publication was peer-reviewed and formally published. If this work was previously peer-reviewed and published, in the cover letter please provide the reason that this work does not constitute dual publication and should be included in the current manuscript.

Reviewers' comments:

Reviewer's Responses to Questions

**Comments to the Author**

1. Is the manuscript technically sound, and do the data support the conclusions?

Reviewer #1: Partly

Reviewer #2: Yes

Reviewer #3: Partly

2. Has the statistical analysis been performed appropriately and rigorously? 

Reviewer #1: I Don't Know

Reviewer #2: Yes

Reviewer #3: I Don't Know

3. Have the authors made all data underlying the findings in their manuscript fully available?

Reviewer #1: No

Reviewer #2: No

Reviewer #3: Yes

4. Is the manuscript presented in an intelligible fashion and written in standard English?

Reviewer #1: No

Reviewer #2: Yes

Reviewer #3: Yes

5. Review Comments to the Author

Reviewer #1: The language used in this paper is outdated. The use of the term "homosexual" is outdated, pathologizing, and has been historically stigmatizing. Then the authors proceed to use an acronym (HOM) on line 12 (it is noted they also do this for heterosexual men) to further stigmatize.

Please see the notes below from GLADD: an American non-governmental media monitoring organization, founded as a protest against defamatory coverage of LGBT people.

Offensive: "homosexual" (n. or adj.)

Preferred: "gay" (adj.); "gay man" or "lesbian" (n.); "gay person/people"

Please use gay or lesbian to describe people attracted to members of the same sex. Because of the clinical history of the word "homosexual," it is aggressively used by anti-gay extremists to suggest that gay people are somehow diseased or psychologically/emotionally disordered – notions discredited by the American Psychological Association and the American Psychiatric Association in the 1970s. Please avoid using "homosexual" except in direct quotes. Please also avoid using "homosexual" as a style variation simply to avoid repeated use of the word "gay." The Associated Press, The New York Times and The Washington Post restrict use of the term "homosexual" (see AP & New York Times Style).

From: https://www.glaad.org/reference/offensive

The authors also seem to be conflating gender and sex with their use of the terms male and men. They need to clearly demonstrate the difference between these two things.

Reviewer #2: 1. Please avoid using the term “homosexual man/men” and “homosexual woman/women.” Instead, use “gay man/men” and “lesbian woman/women.”

2. How are the authors operationally defining “gay community”?

3. Use “scale score reliability” instead of “internal consistency” (i.e., Cronbach’s alpha may be high in situations where item-total correlations are poor).

4. The sample size is quite modest. Do the authors have any idea why recruiting individuals to participate in this study was difficult?

5. 95% confidence intervals should be reported for all Cronbach’s alpha coefficients.

6. For all measures, please provide the possible range of scores and indicate whether higher scores represent more (or less) of the construct of interest.

7. Is the IGCS equally applicable to gay people residing in rural areas (i.e., they may not have access to “gay-affirmative” events)?

8. How was “sexual orientation” measured?

9. On page 15, the authors refer to sexual orientation as an “independent variable.” Technically, this is inaccurate as a person’s sexual orientation cannot be manipulated.

10. What sort of follow-up analysis was used to determine the source of the statistically significant chi-square test? (see Sharpe, D. [2015] "Chi-Square test is statistically significant: Now what?," Practical Assessment, Research, and Evaluation: Vol. 20 , Article 8.

DOI: https://doi.org/10.7275/tbfa-x148

11. Table 2: p values should not be reported as .000

12. Do the authors have any idea why mean scores on the EDS were so low (1.82 and 1.80)? Were there a subset of EDS items that participants reported experiencing more frequently? If so, should those be used grouped and treated as a stand alone indicator of discrimination. Another possibility would be to take the item that was endorsed most often and compare that group to participants who did not experience the specific episode of discrimination.

13. It would be helpful if the authors briefly explained how two group equivalence is determined, and the ways in which equivalence testing and standard hypothesis testing differ.

14. The mean score for the IGCS was 2.91 (near the scale midpoint of 3). Do the authors have any idea why gay men did not report being more invested in the gay male community? (Can lower than expected involvement be linked with COVID-19?)

15. Related to point 14, were there certain items on the IGCS for which gay participants reported high levels of involvement?

16. Did the authors use any quality control items in their survey (e.g., “For this question, please select ‘strongly agree.’”)

17. A diagrammatic representation of the mediation models would be informative.

18. In the summary, the authors should reiterate the value of this manuscript in terms of the incremental advances it offers.

Reviewer #3: The current study examined differences in body image constructs between gay and heterosexual men in German-speaking countries and whether the relationship between sexual orientation and body image facets was mediated by discrimination experiences and gay community involvement. I’m concerned about the rigor of this paper for two reasons: One, that the introduction seems to omit important previous literature, and two, that the methods are somewhat opaque.

General Comments

1. I encourage the authors to adhere to APA standards for bias-free language, especially when speaking about sexual minority individuals: https://apastyle.apa.org/style-grammar-guidelines/bias-free-language/sexual-orientation

In particular, the APA recommends not using “homosexual” and instead using “gay.”

2. Having read the whole paper at this point, I find the title to be misleading as the mediation analyses are only a small part of the overall focus of this paper.

Abstract

3. It’s unclear what “discrimination experiences” consist of for heterosexual men.

4. The mesomorphic ideal consists of both low body fat and high muscularity. As all men are subject to these pressures, I fail to see how differences in facets linked to body fat are explained by the pressures to achieve the mesomorphic ideal.

Introduction

5. There are much more recent citations to assert the fact that gay men are at higher risk of body image disturbance than the Beren paper. I suggest:

He, J., Sun, S., Lin, Z., & Fan, X. (2020). Body dissatisfaction and sexual orientations: A quantitative synthesis of 30 years research findings. Clinical Psychology Review, 81, 101896. https://doi.org/10.1016/j.cpr.2020.101896

6. The authors state that “most studies” did not differentiate their results between cognition and behavior. Are the authors aware of studies that did? If so, these should be cited, and their results should be explained.

7. There are more studies that examine figure rating scales in gay and heterosexual men. See:

Tiggemann, M., Martins, Y., & Kirkbride, A. (2007). Oh to be lean and muscular: Body image ideals in gay and heterosexual men. Psychology of Men & Masculinity, 8(1), 15–24. https://doi.org/10.1037/1524-9220.8.1.15

Meneguzzo, P., Collantoni, E., Bonello, E., Vergine, M., Behrens, S. C., Tenconi, E., & Favaro, A. (2021). The role of sexual orientation in the relationships between body perception, body weight dissatisfaction, physical comparison, and eating psychopathology in the cisgender population. Eating and Weight Disorders - Studies on Anorexia, Bulimia and Obesity, 26(6), 1985–2000. https://doi.org/10.1007/s40519-020-01047-7

8. The authors state that studies “hint at a stronger ED pathology in HOM.” It’s unclear what this means.

9. I suggest an additional citation for discrimination, gay community involvement, eating disorders, and BDD:

Convertino, A. D., Brady, J. P., Albright, C. A., Gonzales IV, M., & Blashill, A. J. (2021). The role of sexual minority stress and community involvement on disordered eating, dysmorphic concerns and appearance- and performance-enhancing drug misuse. Body Image, 36, 53–63. https://doi.org/10.1016/j.bodyim.2020.10.006

Methods

10. Given that it’s an online survey, what kind of data quality checks, checks for repeat participants, etc. were administered? I’m concerned about the quality of the data.

11. In your measures section, please cite any available studies that support the validity of these measures in heterosexual and gay men. Further, any studies that validate the German translations would also be helpful. If none are available, please provide your translation procedures in supplemental materials. I suggest following this paper for guidance:

Swami, V., & Barron, D. (2019). Translation and validation of body image instruments: Challenges, good practice guidelines, and reporting recommendations for test adaptation. Body Image, 31, 204–220. https://doi.org/10.1016/j.bodyim.2018.08.014

12. I would not recommend the Benjamini-Hochberg correction to be separated by “grouping” of outcomes. I’ve never seen it employed this way, and probably increases the false discovery rate. I also wonder how it was employed as there is no mention of this in the results section.

13. The authors test indirect effects/mediation within a cross-sectional design. These indirect effects are not particularly meaningful in such designs. The authors may find the below readings helpful on this topic:

Maxwell, S. E., & Cole, D. A. (2007). Bias in cross-sectional analyses of longitudinal mediation. Psychological Methods, 12(1), 23–44. https://doi.org/10.1037/1082-989X.12.1.23

Maxwell, S. E., Cole, D. A., & Mitchell, M. A. (2011). Bias in Cross-Sectional Analyses of Longitudinal Mediation: Partial and Complete Mediation Under an Autoregressive Model. Multivariate Behavioral Research, 46(5), 816–841. https://doi.org/10.1080/00273171.2011.606716

Results

14. The results section is incredibly difficult to read with so many abbreviations. In general, I would suggest that results are reported in terms of constructs with scale abbreviations in parentheses to facilitate reader experience.

15. In tables 5 and 6, there is a table note that says that significant effects are bolded. This is not true.

Discussion

16. The authors gloss over the finding that drive for thinness was higher among gay men than heterosexual men. What do the authors make of this finding?

17. How can the test neither establish “statistical equality nor statistical differences”? What’s the point of running the test then?

18. I would like to see a citation supporting the statement that “relationships among HOM may be focused on physical relationships rather than on creating a family.” How is this a difference between gay and heterosexual men?

19. Please list additional limitations with the use of perceptual measures as they are numerous. Also, the lack of validation of these measures in the German language and gay and heterosexual men if that validation is lacking.

6. PLOS authors have the option to publish the peer review history of their article (what does this mean?). If published, this will include your full peer review and any attached files.

Reviewer #1: No

Reviewer #2: No

Reviewer #3: **Yes: **Alexandra D. Convertino

---

## [Author Response · Author response to Decision Letter 0]

26 Aug 2022

Response to Editor and Reviewers

Manuscript: Body image disturbance and associated eating disorder and body dysmorphic disorder pathology in gay and heterosexual men: A systematic analyses of cognitive, affective, behavioral und perceptual aspects (PONE-D-21-24159)

Dear Editor and Reviewers,

thank you for your highly important remarks and the chance for clarification and improvement of our manuscript. We did our best to respond to your comments. Please find detailed comments on your remarks and their implementation in the manuscript underneath each notation. The page references refer to the version of our manuscript with track changes.

Editor’s comments to the Author:

Thank you for those very useful templates. We updated the style of our manuscript.

2. We noted in your submission details that a portion of your manuscript may have been presented or published elsewhere. "Henn, Taube, Vocks, & Hartmann, 2019", "BIMTM-MB; Arkenau, Vocks, Taube, Waldorf, & Hartmann, 2020", "Cordes, Vocks, & Hartmann, in press". Please clarify whether this publication was peer-reviewed and formally published. If this work was previously peer-reviewed and published, in the cover letter please provide the reason that this work does not constitute dual publication and should be included in the current manuscript.

Thank you for giving us the opportunity for clarification. We described overlapping use of data and the clear differences between our manuscript and the publications listed above in the cover letter (“Data from the same project on women with different sexual orientations without any data overlap have been peer reviewed and formally published already (Henn, Taube, Vocks, & Hartmann, 2019, DOI: https://doi.org/10.3389/fpsyt.2019.00531; Steinfeld, Hartmann, Waldorf, & Vocks, 2020, DOI: https://doi.org/10.1186/s40337-020-00345-w). Moreover, the sample analyzed in the present paper partly overlaps with a peer reviewed and formally published study by Arkenau, Vocks, Taube, Waldorf and Hartmann (2020, DOI: https://doi.org/10.1002/jclp.22933). In this study, data from male participants (gay or heterosexual) on several measures that were also included in our paper (i. e., Body Appreciation Scale‐2 (BAS‐2; Tylka & Wood‐Barcalow, 2015), Drive for Thinness subscale (DTS) of the Eating Disorder Inventory‐2 (EDI‐2; German version: Paul & Thiel, 2005), Drive for Leanness Scale (DLS; Smolak & Murnen, 2008), Drive for Muscularity Scale (DMS; German version: Waldorf, Cordes, Vocks, & McCreary, 2014), Body Image Disturbance Questionnaire (BIDQ; German version: Hartmann, 2019), Dysmorphic Concern Questionnaire (DCQ; German version: Stangier, Janich, Adam‐Schwebe, Berger, & Wolter, 2003), Eating Disorder Examination‐Questionnaire (EDE‐Q; German version: Hilbert & Tuschen‐Caffier, 2016) were used to validate the Body Image Matrix of Thinness and Muscularity − Male Bodies (BIMTM-MB), a measure that was not included in our study. Hence, data on shared measures were not used to analyze and compare subgroups (i.e., gay vs. heterosexual men) as we did in our manuscript, but only to calculate convergent validity of the BIMTM-MB. Moreover, our sample partly overlaps with a peer reviewed and formally published study that analyzed appearance-related partner preferences in men and women with different sexual orientations (Cordes, Vocks, & Hartmann, 2021, DOI: https://doi.org/10.1007/s10508-021-02087-5). The study’s main focus was the analyses of group differences on the BIMTM-MB. Moreover, correlations of the BIMTM-MB and the DMS, EDI-2 and the EDE-Q were calculated to analyse associations of appearance-related partner preferences and body image between groups. Hence, data on the DMS, EDI-2 and EDE-Q were not used to analyze and compare group differences on body image as we did in our study. To conclude, our manuscript does not constitute dual publication. For verification purpose please see all open access studies via the DOIs listed.“). 

Upon your advice, we made data from this study available by uploading the minimal anonymized data set necessary to replicate our study findings as a public repository (Open Science Framework). See: DOI 10.17605/OSF.IO/KFYZ7 

As we do not have any supporting information apart from our research data that has been made publicly available in a repository (see above), we assume that this comment does not apply. Please advise if we misunderstood your comment.

Reviewers' comments to the Author:

Reviewer #1: 

The language used in this paper is outdated. The use of the term "homosexual" is outdated, pathologizing, and has been historically stigmatizing. Then the authors proceed to use an acronym (HOM) on line 12 (it is noted they also do this for heterosexual men) to further stigmatize.

Please see the notes below from GLADD: an American non-governmental media monitoring organization, founded as a protest against defamatory coverage of LGBT people.

Offensive: "homosexual" (n. or adj.)

Preferred: "gay" (adj.); "gay man" or "lesbian" (n.); "gay person/people"

Please use gay or lesbian to describe people attracted to members of the same sex. Because of the clinical history of the word "homosexual," it is aggressively used by anti-gay extremists to suggest that gay people are somehow diseased or psychologically/emotionally disordered – notions discredited by the American Psychological Association and the American Psychiatric Association in the 1970s. Please avoid using "homosexual" except in direct quotes. Please also avoid using "homosexual" as a style variation simply to avoid repeated use of the word "gay." The Associated Press, The New York Times and The Washington Post restrict use of the term "homosexual" (see AP & New York Times Style).

From: https://www.glaad.org/reference/offensive

Thank you for this extremely helpful comment! As non-native English speakers, we have not been aware of this and of course replaced the word “homosexual man” with “gay man” throughout the manuscript. Also, we deleted all acronyms referring to gay or heterosexual men.

The authors also seem to be conflating gender and sex with their use of the terms male and men. They need to clearly demonstrate the difference between these two things.

Thank you for giving us the opportunity for clarification. The German language does not differentiate between sex (“Geschlecht”) and gender (“Geschlecht” as well). In our self-report study, we asked participants how they personally refer to themselves in terms of their “Geschlecht”, which we then interpreted as their gender, not their sex. This is why we explicitly stated in the manuscript: “data were gathered on [participant’s] age, gender, nationality […]” (p.20, l. 446 ff.). In the manuscript, we use the term male as the adjective to man for descriptive purposes (e.g. “the male gay community”). We added this information to the manuscript (“Also, we use male as the adjective of man and not to differentiate between gender and sex.”) (p. 12, l. 259 ff.). Please advise if we misunderstood your comment, we are happy to implement further suggestions.

Reviewer #2: 

1. Please avoid using the term “homosexual man/men” and “homosexual woman/women.” Instead, use “gay man/men” and “lesbian woman/women.”

Thank you for this important comment. As stated above, we replaced the word “homosexual” with the word “gay” throughout the manuscript. 

2. How are the authors operationally defining “gay community”?

Thank you for giving us the opportunity for clarification. First of all, we define “community” as a group of people that has a shared characteristic (Holt, 2011), and in terms of the gay community this shared characteristic is being gay. We added this to our manuscript as follows: “To clarify, we defined gay community as a group of people with the shared characteristic of being gay, and gay community involvement as engaging with other members of the gay community (i. e. gay men) and active participation in gay community spaces and activities, such as attending pride events, visiting a gay bar or reading a gay newspaper [51,57].” (p. 12, l. 255 ff.)

3. Use “scale score reliability” instead of “internal consistency” (i.e., Cronbach’s alpha may be high in situations where item-total correlations are poor).

Thank you for giving us the opportunity to improve our methods section. As we were not sure how to interpret your comment, we sought advise with our statistics expert. He agreed that Cronbach’s alpha is a controversially discussed measure for the reliability of a test, but unfortunately also remained unsure what alternative measure you would suggest. In case you insist on a different measure than Cronbach’s alpha, could you please specify your comment? Thank you in advance. As described for comment 5, we added 95% confidence intervals for all Cronbach’s alpha coefficients.

4. The sample size is quite modest. Do the authors have any idea why recruiting individuals to participate in this study was difficult?

Thank you for giving us the opportunity for clarification. Please keep in mind that we only recruited participants in German-speaking countries, which means that the population that could be targeted was much smaller than if we had conducted our study in a language that is more frequently spoken, like English or Spanish. Moreover, we targeted a quite narrow group of participants (first and foremost people with a sexual orientation other than heterosexuality). Also, we were transparent about the topic of the survey (sexual orientation and body image), which might have discouraged some potential participants from participation due to the ongoing stigma surrounding sexual orientation. Furthermore, in Germany we don’t have access to data collection tools like Amazon Mechanical Turk or Taskrabbit. Generally, budget to compensate study participation is usually small, so researchers rely on intrinsic motivation of potential participants. In our study, participants only had the opportunity to take part in a lottery to win an online shopping voucher (1 out of 10, worth 20 Euros) but no guaranteed compensation was payed. Moreover, budget for study promotion is usually small, which is why we had to limit our advertising activity to university e-mail distribution lists, posters, flyers, voluntary press releases on lesbian, gay, bisexual, and transgender (LGBTQ) websites and Facebook groups. Taken together, the sample size 

(n = 838 for the broader project and n = 262 that identified as men and could be included in our study), met our expectations, especially compared to other surveys on sexual orientation in German-speaking countries (e. g. Legenbauer et al. (2009), n = 305 participants from which n = 143 identified as men; Siever (1994), n = 250 participants from which n = 122 identified as men). Also, a priori power analyses indicated that we would need a sample size of n = 111 to detect a medium-sized effect (p = 0.3).

5. 95% confidence intervals should be reported for all Cronbach’s alpha coefficients.

Thank you for the comment, we added 95% confidence intervals for all Cronbach’s alpha coefficients, e. g. p. 15, l. 316: “α = .92, 95% CI [0.91, 0.94]”

6. For all measures, please provide the possible range of scores and indicate whether higher scores represent more (or less) of the construct of interest.

Thanks for the suggestion. We added the possible range of scores und how to interpret scores where they have not already been stated, e. g. p. 15, l. 329: “Larger discrepancies indicate greater perceptual body image disturbance.”.

7. Is the IGCS equally applicable to gay people residing in rural areas (i.e., they may not have access to “gay-affirmative” events)?

Thanks for this interesting question. The IGCS does not make any comments on that. However, especially in a small country like Germany, bigger cities or more progressive areas, where the probability for such activities might be bigger, can usually be accessed in 30 to 60 minutes, even when living on the countryside or in conservative areas. Also, the majority of questions on the IGCS does not depend on participant’s location. Only one item specifically asks about attendance to gay affirmative-events. Therefore, we do not expect the item to have massively biased our results.

8. How was “sexual orientation” measured?

Sexual orientation was measured via self-report of the participants. They were able to choose from a range of different categories of sexual orientation (although we explicitly acknowledged that sexual orientation is a continuum), or type in their sexual orientation in a text field if none of the categories met their sexual orientation. We clarified this in the manuscript as follows: “Sexual orientation was measured via self-report of the participants. They were able to choose from a range of different categories of sexual orientations (gay, lesbian, heterosexual, bisexual, pansexual, polysexual, asexual), although we explicitly acknowledged that sexual orientation is a continuum. If none of the categories met their sexual orientation, participants could type in their sexual orientation in a text field.” (p. 20, l. 442 ff).

9. On page 15, the authors refer to sexual orientation as an “independent variable.” Technically, this is inaccurate as a person’s sexual orientation cannot be manipulated.

Thanks for making us aware of this. As we deleted the mediation analyses as suggested by Reviewer #3, the comment has become obsolete.

10. What sort of follow-up analysis was used to determine the source of the statistically significant chi-square test? (see Sharpe, D. [2015] "Chi-Square test is statistically significant: Now what?," Practical Assessment, Research, and Evaluation: Vol. 20 , Article 8.

DOI: https://doi.org/10.7275/tbfa-x148

Thank you for this important suggestion. We added the following follow-up analysis of the chi-square test: “In case of a significant χ2 test, adjusted residuals were calculated and checked to locate the source of the significance. An adjusted residual with an absolute value that exceeded +/- 1.96 indicated lack of fit of the Null-hypothesis, i.e. significance [78].” (p. 21, l. 456 ff.) The result of the follow-up analyses, as stated in the manuscript, was as follows: “The observation of the adjusted residuals suggested that the rejection of the Null-hypothesis [regarding highest educational attainment] resulted as, compared to heterosexual men, a larger number of gay men had no higher-track secondary school qualification than statistically expected.” (p. 23, l. 505 ff.)

11. Table 2: p values should not be reported as .000

We changed the p-value to “< .001” (see Table 2) 

12. Do the authors have any idea why mean scores on the EDS were so low (1.82 and 1.80)? Were there a subset of EDS items that participants reported experiencing more frequently? If so, should those be used grouped and treated as a standalone indicator of discrimination. Another possibility would be to take the item that was endorsed most often and compare that group to participants who did not experience the specific episode of discrimination.

Thank you for the interesting question that we were happy to follow up on. Detailed analysis of the dataset led to the following results: First of all, possible item scores on the EDS range from 1 (“Never [have I experienced such a discrimination experience]”) to 6 (“Almost every day [do I experience such a discrimination experience]”). In our study, mean scores of the individual items of the EDS ranged from M = 0.03 to M = 2.18 in the group of gay men and from 0.05 to 2.45 in the group of heterosexual men. The median throughout items in the group of gay men was 1.83 and in the group of heterosexual men 1.67. The three items with the highest mean scores were exactly the same in both groups (No 1. “People pretending to be better than you”, gay men: M = 2.39, SD = 1.28, heterosexual men M = 2.45, SD = 1.40); 

No 2. “You are treated less polite than other people”, gay men: M = 2.18, SD = 1.25, heterosexual men M = 2.07, SD = 1.28; No 3. “You are treated with less respect than other people”, gay men: M = 2.18, SD = 1.10, heterosexual men M = 1.99, SD = 1.21). This indicates an overall low score of possible discrimination experiences throughout all items in both groups. So overall, our sample does not experience any frequent discrimination experiences, and the two subgroups did not differ in this regard. Therefore, further analyses of single items did not appear promising.

13. It would be helpful if the authors briefly explained how two group equivalence is determined, and the ways in which equivalence testing and standard hypothesis testing differ.

Thank you for the suggestion. We described the procedure and the difference to standard 

t-testing in more detail: “To test for equivalence of groups, two one-sided t-tests (TOST) were calculated. This test is a variation of the standard one-sided t-test, that examines whether the hypothesis that the difference between two groups is zero can be rejected. The TOST, however, examine whether the hypothesis that the difference between groups is meaningful (i. e., at least as extreme as the smallest effect size of interest) can be rejected. The smallest effect size of interest was set using established benchmarks [81], namely at d = 0.2, which represents a trivially small effect size [80]. Groups are considered equivalent when both of the two one-sided t-tests are statistically significant. In the case of heterogeneity of variance, Welch’s tests were employed.” (p. 21, l. 467).

14. The mean score for the IGCS was 2.91 (near the scale midpoint of 3). Do the authors have any idea why gay men did not report being more invested in the gay male community? (Can lower than expected involvement be linked with COVID-19?)

That is an interesting question. The COVID-19 pandemic can not have influenced the results as the study was conducted from 2017 to 2018. Moreover, the authors of the IGCS report a mean score of M = 3.09 in a male gay sample in the validation of the IGCS, which is close to the score that we found in our study. Therefore, we interpret our gay sample to be ordinarily engaged in the gay community. 

15. Related to point 14, were there certain items on the IGCS for which gay participants reported high levels of involvement?

To clarify, the IGCS does not only measure involvement, but also identification with the gay community. In the manuscript we discussed this as a possible reason for the non-significant association of the IGCS and body image disturbances in our gay sample, as predominantly involvement with the gay community “is suspected to convey specific unrealistic body ideals that contribute to body discontent” (p. 41, l. 790 ff). As the authors of the IGCS do not clearly define which items refer to involvement and which items refer to identification with the gay community, we were not able to calculate subscales to test for this hypothesis. We added this information to the manuscript (“Since the authors of the IGCS do not clearly define which items refer to involvement and which items refer to identification with the gay community, we were not able to calculate subscales to test for this hypothesis”, p. 41, l. 793 ff) To come back to your question, mean scores of the individual items of the IGCS ranged from M = 1.65 to M = 3.72. The two items with the lowest scores refer to the attendance of LGBTQ events 

(M = 1.65, SD = 1.04) or LGBTQ bars/discotheques (M = 1.83, SD = 0.94). The two items with the highest scores are “Being attracted to men is important to my sense of who I am” (M = 3.72, SD = 1.13) and “How often do you read a gay oriented (online-)paper or magazine, or visit websites that focus on being gay?” (M = 3.46, SD = 1.39). 

16. Did the authors use any quality control items in their survey (e.g., “For this question, please select ‘strongly agree.’”) 

Thanks for this valid question. No, we did not use any quality control items. As stated above, we did not use any data collection tools like Amazon Mechanical Turk or Taskrabbit that increase chances for reduced data quality due to the incentive system of the tools. Participants did not get any compensation for participation but only had the opportunity to take part in a lottery to win an online shopping voucher (1 out of 10, worth 20 Euros). Hence, we expected participants to take part in the survey due to an intrinsic motivation to support research in the field of sexual orientation and body image. Therefore, we expected participants to answer sincerely and did not see an urgent need to implement quality control items. Nevertheless, we visually checked data for conspicuous answering patterns but did not find any anomalies. We added that information in the manuscript for clarification (“Further visual observation of data did not detect any conspicuous answering patterns”, p. 13, l. 277 ff.; “Upon completion, participants were given the opportunity to take part in a lottery to win an online shopping voucher (1 out of 10, worth 20 Euros). No further compensation for participation was payed.”, p. 15, l. 307 ff.). Nevertheless, to further improve data quality, we will consider implementing data control items in our next survey.

17. A diagrammatic representation of the mediation models would be informative.

Thanks for this suggestion. However, we deleted the mediation analyses upon recommendation by Reviewer #3.

18. In the summary, the authors should reiterate the value of this manuscript in terms of the incremental advances it offers.

Thanks for the comment. We made the following changes: “The present study contributes to the extensive, yet narrow research on body image in gay men by systematically examining multiple dimensions of body image disturbance in gay and heterosexual men as well as associated pathologies, including the under-investigated BDD pathology. Overall, the results suggest that gay men not only show more body dissatisfaction than heterosexual men but a significantly higher multidimensional body image disturbance affecting cognitions, emotions, behaviors, and perception. This might be especially true for facets linked to body weight and thinness, suggesting the dilemma of a paradoxically thin, yet muscular body ideal for gay men, that might not even be dissolved in a lean body. In accordance with that, gay man also showed significantly higher ED and BDD pathology for which body image disturbance poses an eminently relevant risk factor [10,11]. However, differences in body image might not be associated with the frequency of everyday discrimination experiences and cross-subgroup involvement with the gay community.” (p. 43, l. 844 ff.). Also, we changed the order of the introduction to stress the focus of the paper and the research gap it aims to close.

Reviewer #3: 

The current study examined differences in body image constructs between gay and heterosexual men in German-speaking countries and whether the relationship between sexual orientation and body image facets was mediated by discrimination experiences and gay community involvement. I’m concerned about the rigor of this paper for two reasons: One, that the introduction seems to omit important previous literature, and two, that the methods are somewhat opaque.

General Comments

1. I encourage the authors to adhere to APA standards for bias-free language, especially when speaking about sexual minority individuals: https://apastyle.apa.org/style-grammar-guidelines/bias-free-language/sexual-orientation

In particular, the APA recommends not using “homosexual” and instead using “gay.”

Thanks a lot for making us aware of that! We changed the word “homosexual” to “gay”.

2. Having read the whole paper at this point, I find the title to be misleading as the mediation analyses are only a small part of the overall focus of this paper.

Thank you for this valid comment. As we deleted the mediation analyses (see comment 13), we changed the title to “Body image disturbance and associated eating disorder and body dysmorphic disorder pathology in gay and heterosexual men: A systematic analyses of cognitive, affective, behavioral und perceptual aspects” to better fit the focus of the manuscript.

Abstract

3. It’s unclear what “discrimination experiences” consist of for heterosexual men.

Thanks for making us aware of that. The Everyday Discrimination Scale (EDS) used in the study does not only measure discrimination experiences specifically linked to sexual orientation. We expanded on that in the method section of our manuscript (“The tenth item asks about the specific self-suspected reason for discrimination, like age, nationality, or sexual orientation. Therefore, the EDS does not only apply to discrimination experiences based on sexual orientation.”, p. 20, l. 435 ff.). Also, we added the prefix “general everyday discrimination experiences” in the abstract to make this clearer (e. g. p. 3, l. 27).

4. The mesomorphic ideal consists of both low body fat and high muscularity. As all men are subject to these pressures, I fail to see how differences in facets linked to body fat are explained by the pressures to achieve the mesomorphic ideal.

Thanks for your comment. By referring to the mesomorphic ideal, we tried to explain why we found no difference between our gay and heterosexual sample in terms of the cognitive and behavioral drive for muscularity. You are right, our result that gay men had an elevated drive to be thin (not to be confused with lean, which refers to an athletic physique with low body fat and visible muscularity), cannot be explained by the mesomorphic ideal. To prevent confusion, we deleted this aspect from the abstract and described our interpretation of the findings regarding the body ideal in gay vs. heterosexual men in the discussion of the manuscript as follows: “The results [of both a drive to be thin and to be muscular in gay men] contradict findings proposing that the drive for a thin body is a “female” body image issue [85,87,88] and underlines the paradox of a highly light-weight, yet wide and muscular body ideal for gay men [17]. This rather unattainable body ideal might cause a dilemma influencing the highly elevated body dissatisfaction in gay men.” (p., 36, l. 680ff). Also, we described the differentiation between drive for thinness, leanness and muscularity in the introduction in more detail: “In more detail, gay men seem to strive more strongly for a thin body (i. e., low body weight) [17,21,22], although some studies have reported similar levels of drive for thinness between gay and heterosexual men [20]. At the same time, there appears to be no difference between gay and heterosexual men in drive for muscularity (i. e., a muscular, broad physique) [19,20]. However, previous studies did not differentiate their results in terms of muscle-related cognitions and muscle-related behaviors, with the latter being better categorized as part of behavioral body image disturbance [3]. Concerning the drive for a lean body (i.e., a trained, tight physique), no comparative study exists.” (p. 6, l. 104 ff.)

Introduction

5. There are much more recent citations to assert the fact that gay men are at higher risk of body image disturbance than the Beren paper. I suggest:

He, J., Sun, S., Lin, Z., & Fan, X. (2020). Body dissatisfaction and sexual orientations: A quantitative synthesis of 30 years research findings. Clinical Psychology Review, 81, 101896. https://doi.org/10.1016/j.cpr.2020.101896

Thank you for the suggestion. We implemented this valuable citation as follows: “For example, a quantitative synthesis of 30 years of research findings on body dissatisfaction and sexual orientation found significantly higher body dissatisfaction in sexual minority men than in heterosexual men [15], that might be similarly high [16] or even higher [17] than in heterosexual women.” (p. 5, l. 772 ff.).

6. The authors state that “most studies” did not differentiate their results between cognition and behavior. Are the authors aware of studies that did? If so, these should be cited, and their results should be explained.

Thanks for pointing that out. Indeed, to our knowledge none of the studies that compared gay and straight men regarding drive for muscularity differentiated their results in terms of muscle-related cognitions and muscle-related behaviors. Therefore, we erased the word “most” and apologize for the confusion.

7. There are more studies that examine figure rating scales in gay and heterosexual men. See:

Tiggemann, M., Martins, Y., & Kirkbride, A. (2007). Oh to be lean and muscular: Body image ideals in gay and heterosexual men. Psychology of Men & Masculinity, 8(1), 15–24. https://doi.org/10.1037/1524-9220.8.1.15

Meneguzzo, P., Collantoni, E., Bonello, E., Vergine, M., Behrens, S. C., Tenconi, E., & Favaro, A. (2021). The role of sexual orientation in the relationships between body perception, body weight dissatisfaction, physical comparison, and eating psychopathology in the cisgender population. Eating and Weight Disorders - Studies on Anorexia, Bulimia and Obesity, 26(6), 1985–2000. https://doi.org/10.1007/s40519-020-01047-7

Thanks for your valuable suggestions. We added both studies to our reference list. The paper by Tiggemann et al. (2007) conceptualizes the discrepancy between one’s self-rated current and ideal figure as a distortion of perception (like we did), however Meneguzzo et al. (2021) interpreted the discrepancy as body weight dissatisfaction. Therefore, we listed the latter study in our limitations section, where we noted that “the discrepancy between self-rated current and ideal body could also be interpreted as perceptual body discontent [102,103] and therefore allocated to the cognitive-affective facet of body image disturbance” (p. 42, l. 826 ff.).

8. The authors state that studies “hint at a stronger ED pathology in HOM.” It’s unclear what this means.

Thanks for giving us the opportunity for clarification. We added the following information: “Studies have indeed found higher prevalence rates of EDs in gay men compared to heterosexual men [32,33] as well as a more pronounced ED pathology [e.g., 20,34]. Also, gay men appear to exhibit more severe ED symptoms such as binge eating [34], purging behavior [35], restrictive eating [36,31], and taking weight-reducing supplements [35]. (p. 7, l. 140 ff.)

9. I suggest an additional citation for discrimination, gay community involvement, eating disorders, and BDD:

Convertino, A. D., Brady, J. P., Albright, C. A., Gonzales IV, M., & Blashill, A. J. (2021). The role of sexual minority stress and community involvement on disordered eating, dysmorphic concerns and appearance- and performance-enhancing drug misuse. Body Image, 36, 53–63. https://doi.org/10.1016/j.bodyim.2020.10.006

Thank you for giving us the opportunity to update our introduction section. We included the paper and its further implications as follows: “Besides discrimination, it is assumed that pressure from within the gay community to be attractive and muscular might also contribute to elevated body image concerns among gay men [45,46]. According to the intraminority stress theory [45], masculinity and attractiveness are means to gain status among the gay community, leading to appearance-based comparisons and competition with other community members, as well as pressure to conform to an attractive and muscular body ideal. This pressure is said to be further reinforced as gay and bisexual men usually rely on other men from within their sexual minority community for sexual and social relationships [45]. Therefore, sexual minority community involvement has been linked to negative body image outcomes among sexual minority men [46]. For example, Hospers and Jansen [48] found increased pressure to conform to appearance standards in order to attract sexual partners within the gay community. Furthermore, Convertino et al. [49] reported elevated rates of disordered body image behaviors and concerns depending on gay community involvement.” (p. 9, l. 175 ff.).

Methods

10. Given that it’s an online survey, what kind of data quality checks, checks for repeat participants, etc. were administered? I’m concerned about the quality of the data.

Thanks for this valid question. As we pointed out above, we did not use any quality control items. Participants did not get any compensation for participation in the 35-minutes-long survey, but only had the opportunity to take part in a lottery to win an online shopping voucher (1 out of 10, worth 20 Euros). Also, we did not use any data collection tools like Amazon Mechanical Turk or Taskrabbit that increase chances for reduced data quality due to the incentive system of those tools. Hence, we expected participants to take part in the survey due to an intrinsic motivation to support research in the field of sexual orientation and body image. Therefore, we expected participants to answer sincerely and did not see an urgent need to implement quality control items. Nevertheless, we visually checked data for conspicuous answering patterns but did not find any anomalies. We added that information in the manuscript for clarification (“Further visual observation of data did not detect any conspicuous answering patterns”, p. 13, l. 277; “Upon completion, participants were given the opportunity to take part in a lottery to win an online shopping voucher (1 out of 10, worth 20 Euros). No further compensation for participation was payed.”, p. 15, l. 307 ff.). Nevertheless, to further improve data quality, we will consider implementing data control items in our next survey.

11. In your measures section, please cite any available studies that support the validity of these measures in heterosexual and gay men. Further, any studies that validate the German translations would also be helpful. If none are available, please provide your translation procedures in supplemental materials. I suggest following this paper for guidance:

Swami, V., & Barron, D. (2019). Translation and validation of body image instruments: Challenges, good practice guidelines, and reporting recommendations for test adaptation. Body Image, 31, 204–220. https://doi.org/10.1016/j.bodyim.2018.08.014

Thanks a lot for your suggestions. If available, we provided studies that support the validity of our measures in sexual minority men: 

• Body Appreciation Scale-2: “The BAS-2 has been validated for the use with a male and female sexual minority sample could [65]” (p. 15, l. 318 ff.)

• Bodybuilder Image Grid-Original (BIG-O): “The BIG-O is validated for the use in a male sample [28] and has been used in studies investigating body image in gay men (e.g., [31]).” (p. 16, l. 332)

• Body Image Coping Strategies Inventory: “The BICSI is validated for the use in a male sample [60] but has not yet been validated or used for a gay sample.” (p. 16, l. 344 ff.)

• Body Image Disturbance Questionnaire: “The BIDQ is validated for the use in a male sample [66] but has not yet been validated on or used for a gay sample.” (p. 17, l. 357)

• Dysmorphic Concern Questionnaire: “The DCQ has been validated for the use with a sexual minority sample [69]” (p. 17, l. 364)

• Drive for Leanness Scale: “The DLS is validated for the use in a male sample [71] but has not yet been validated on or used for a gay sample.” (p. 17, l. 372)

• Drive for Muscularity Scale: “The DMS has been validated for the use with a male sexual minority sample [72]” (p. 18, l. 387 ff.)

• Drive for Thinness Scale: “The DTS has not yet been validated for the use in a gay sample but has been used in studies investigating body image in gay men (e.g., [20])”

• Eating Disorder Examination-Questionnaire: “The EDE-Q has been validated for the use with a male sexual minority sample [75].” (p. 19, l. 410)

• Gender-Neutral Body Checking Questionnaire: “The GNBCQ has been validated on a male, but not on a sexual minority sample [61].”(p. 19, l. 419 ff.)

• The Everyday Discrimination Scale: “The EDS has not yet been validated or used on a gay sample.” (p. 20, l. 441)

Furthermore, as we could not provide validation studies on a sexual minority sample for all instruments we used, we calculated internal consistency of our measures for both groups separately and added this information to our measures section. With the exception of the Body Image Coping Strategies Inventory, subscale Avoidance, all scales yielded high to excellent internal consistency for the gay and heterosexual subgroup, respectively. The subscale Avoidance yielded acceptable internal consistency for both subgroups, however it also only yielded acceptable internal consistency in the original validation study.

Regarding our translation process: we used Brislin`s (1970) back-translation to translate measures where no German version was available. A bilingual translator (German – English) blindly translated (i.e., forward-translated) the original English language measure, including instructions and response categories, to German. Then, a second bilingual translator independently back-translated the instrument from German to the original English language. Afterwards, the two language versions of the measurement (i.e., the original English and back-translated English versions) were compared for conceptual, item, semantic, and operational equivalence. If discrepancies occurred, another translator tried to retranslate the relevant item. This process was continued until all bilingual translators agreed that the two versions of the instrument are identical. We added the translation procedure to our methods section (p., 14, l. 293 ff)

The guidelines presented in the Paper by Swami and Barron (2019) present a valuable a priori procedure that we unfortunately did not follow. We stated this as a limitation of our study in the discussion section: “Also, as there were no previously validated German language versions of the Body Appreciation Scale - 2 [64], the Body Image Coping Strategies Inventory [60], the Drive for Leanness Scale [70], the Gender Neutral Body checking Questionnaire [61], the Identification with Gay Community Scale [57] and The Everyday Discrimination Scale [62] we had to translate those by ourselves via back-translation [63]. We did not conduct a comprehensive validation procedure of the translated measures, but the internal consistencies of the translated measures merely differ from those of the originals.” (p. 43, l. 836 ff.). 

12. I would not recommend the Benjamini-Hochberg correction to be separated by “grouping” of outcomes. I’ve never seen it employed this way, and probably increases the false discovery rate. I also wonder how it was employed as there is no mention of this in the results section.

Thank you for your comment that we are happy to follow. We performed the Benjamini-Hochberg correction without grouping of outcomes and reported the newly adjusted p-values in Table 2. Also, in the table notes we added that the reported p-values are Benjamini-Hochberg adjusted.

13. The authors test indirect effects/mediation within a cross-sectional design. These indirect effects are not particularly meaningful in such designs. The authors may find the below readings helpful on this topic:

Maxwell, S. E., & Cole, D. A. (2007). Bias in cross-sectional analyses of longitudinal mediation. Psychological Methods, 12(1), 23–44. https://doi.org/10.1037/1082-989X.12.1.23

Maxwell, S. E., Cole, D. A., & Mitchell, M. A. (2011). Bias in Cross-Sectional Analyses of Longitudinal Mediation: Partial and Complete Mediation Under an Autoregressive Model. Multivariate Behavioral Research, 46(5), 816–841. https://doi.org/10.1080/00273171.2011.606716

Thank you for your comment. You are right, as we did not use longitudinal data, mediation analyses might be biased. After careful consideration, we deleted to mediation analyses from the manuscript.

Results

14. The results section is incredibly difficult to read with so many abbreviations. In general, I would suggest that results are reported in terms of constructs with scale abbreviations in parentheses to facilitate reader experience.

Thank you for making us aware of this. We put the abbreviations in parentheses after naming the actual construct (e. g. “total eating disorder pathology (EDE-Q total score)”, p. 28, l. 564)

15. In tables 5 and 6, there is a table note that says that significant effects are bolded. This is not true.

Thank you for your comment. You are correct, in table 5 and 6 no effects are marked in bold as there have not been any significant effects. We understand that this might cause confusion and deleted the note (see Table 5)

Discussion

16. The authors gloss over the finding that drive for thinness was higher among gay men than heterosexual men. What do the authors make of this finding?

We added the following discussion of the above-mentioned finding: “The results contradict findings proposing that the drive for a thin body is a “female” body image issue [85,87,88] and underlines the paradox of a highly light-weight, yet wide and muscular body ideal for gay men [17]. This rather unattainable body ideal might cause a dilemma influencing the highly elevated body dissatisfaction in gay men.” (p. 36, l. 687 ff.)

17. How can the test neither establish “statistical equality nor statistical differences”? What’s the point of running the test then?

Thanks for giving us the opportunity for clarification. The two one-sided t-tests (TOST) that we ran to check for statistical equivalence of groups is a variation of the standard one-sided t-test. The t-test examines whether the hypothesis that the difference between two groups is zero can be rejected. The TOST, however, examines whether the hypothesis that the difference between groups is meaningful (i. e. at least as extreme as the smallest effect size of interest) can be rejected. In our study, firstly we ran a TOST to check for equivalence of groups. After the TOST was non-significant, indicating that groups were not statistically equivalent, we ran a one-sided t-test to check if groups significantly differed from each other. Again, this test was not significant, indicating that groups were neither significantly equal, nor significantly different from each other. This indicates that the difference between the two groups is somewhere between zero and the smallest effect size of interest. As such, it is not possible to sufficiently interpret those results. We described this procedure in more detail in the method section and in the results section (“To test for equivalence of groups, two one-sided t-tests (TOST) were calculated. The test is a variation of the standard one-sided t-test, that examines whether the hypothesis that the difference between two groups is zero can be rejected. The TOST, however, examine whether the hypothesis that the difference between groups is meaningful (i. e. at least as extreme as the smallest effect size of interest) can be rejected. The smallest effect size of interest was set using established benchmarks [81], namely at d = 0.2, which represents a trivially small effect size [80]. Groups are considered equivalent when both of the two one-sided t-tests are statistically significant. In case the TOST was non-significant, indicating that groups are not statistically equivalent, a one-sided t-test was run to check if groups significantly differed from each other.” p. 21, l. 468.; “The equivalence tests (TOSTs) as well as the null-hypothesis tests (one-sided t-tests) regarding the discrepancy between current and ideal muscularity (BIG-O subscale) and muscle-related behavior (DMS subscale) were both non-significant., This indicates that groups were neither statistically equal, nor significantly different from each other. Hence, the difference between the two groups was somewhere between zero and the smallest effect size of interest that was previously set. As such, it is not possible to sufficiently interpret those results.”, p. 25, l. 528 ff.). 

Also, we described the problem as a limitation in our discussion section: “Lastly, we found neither statistical equivalence nor significant differences between the groups regarding behavioral drive for muscularity and discrepancy between self-rated current and ideal body fat. This might be due to our method of determining the smallest effect size of interest, which was based on established benchmarks (Cohen`s effect size conventions [80]), and not on related studies in the literature or individual, empirical considerations [81]. As such, it is not possible to sufficiently interpret the results, p. 43, l. 844“). 

18. I would like to see a citation supporting the statement that “relationships among HOM may be focused on physical relationships rather than on creating a family.” How is this a difference between gay and heterosexual men?

Thanks for raising this question. We described our point in more detail as follows: “Lastly, it was argued that intracommunity pressure to conform to a certain body ideal stems from the wish to attract other members of the community for sexual and social relationships [45]. However, 42% of gay men in our study were in a committed relationship, which was a similar quantity as in heterosexual men. This may have lessened the pressure from the male gay community on our gay sample to stay attractive.” (p., 42, l. 811 ff.)

19. Please list additional limitations with the use of perceptual measures as they are numerous. Also, the lack of validation of these measures in the German language and gay and heterosexual men if that validation is lacking. 

Thank you for your comment. We added the following limitations for the use of figural drawing scales to measure perceptual body image distortion: “Regarding the use of a figural drawing scale to measure perceptual body image distortion, general limitations of these measures include that scales only display hand-drawn und therefore less detailed body images, that are based on an artist’s subjective belief of varying body weight and muscularity. Also, those measures only depict a limited set of varying bodies, while in reality body shape is a continuous variable [58]. Those aspects might have limited the validity of our results.” (p. 42, l. 825). As stated for comment 11, we also added the lack of proper validation of the back-and-forth translated measures as a limitation of our study.

---

## [Decision Letter · Decision Letter 1]

21 Sep 2022

PONE-D-21-24159R1Body image disturbance and associated eating disorder and body dysmorphic disorder pathology in gay and heterosexual men: A systematic analyses of cognitive, affective, behavioral und perceptual aspectsPLOS ONE

Dear Dr. Schmidt,

Thank you for submitting your manuscript to PLOS ONE. After careful consideration, we feel that it has merit but does not fully meet PLOS ONE’s publication criteria as it currently stands. Therefore, we invite you to submit a revised version of the manuscript that addresses the points raised during the review process.

The authors adequately responded to the Reviewers' comments.However, minor revisions are necessary for the presentation of the study.

See the comments and respond to them appropriately.

We look forward to receiving your revised manuscript.

Kind regards,

Masaki Mogi

Academic Editor

PLOS ONE

Journal Requirements:

Reviewers' comments:

Reviewer's Responses to Questions

**Comments to the Author**

1. If the authors have adequately addressed your comments raised in a previous round of review and you feel that this manuscript is now acceptable for publication, you may indicate that here to bypass the “Comments to the Author” section, enter your conflict of interest statement in the “Confidential to Editor” section, and submit your "Accept" recommendation.

Reviewer #3: (No Response)

2. Is the manuscript technically sound, and do the data support the conclusions?

Reviewer #3: Yes

3. Has the statistical analysis been performed appropriately and rigorously? 

Reviewer #3: Yes

4. Have the authors made all data underlying the findings in their manuscript fully available?

Reviewer #3: Yes

5. Is the manuscript presented in an intelligible fashion and written in standard English?

Reviewer #3: Yes

6. Review Comments to the Author

Reviewer #3: Thank you for the opportunity to read the revised manuscript entitled “Body image disturbance and associated eating disorder and body dysmorphic disorder pathology in gay and heterosexual men: A systematic analyses of cognitive, affective, behavioral und perceptual aspects.” I find that the paper has been much improved through the review process and am impressed by the authors responsiveness to reviews.

General comments:

1. To continue the commentary on gender and sex differentiation in this paper, I appreciate the consideration that the authors have already paid, and challenges related to language differences across English and German. I would also still encourage the authors to avoid the term “male” when possible, even as an adjective, due to specific language differentiation in English. Therefore, I would encourage the removal of the sentence: “Also, we use male as the adjective of man and not to differentiate between gender and sex” and instead rephrase to avoid using male. (E.g., “…Buhlmann and colleagues found that 27% of MEN IN THE STUDY reported at least one body-related concern…” as opposed to “…Buhlmann and colleagues found that 27% of MALE PARTICIPANTS reported at least one body-related concern…”

Abstract

2. “Extensive, yet narrow” is a confusing phrase that the authors use multiple times throughout the paper now. Narrow is an antonym of extensive and therefore, this is somewhat oblique in meaning. I would encourage the authors to use a different phrase. If the literature is quantitively large, but narrow in scope, that is the phrase I would use.

Introduction

3. On page 4, line 81-82, the authors state that no studies have compared drive for leanness between gay and heterosexual men. This is not strictly true, although their analysis do make interpretation difficult. See:

Strübel, J., & Petrie, T. A. (2019). Appearance and performance enhancing drug usage and psychological well-being in gay and heterosexual men. Psychology & Sexuality, 10(2), 132–148. https://doi.org/10.1080/19419899.2019.1574879

4. I have a bit of a quibble with the assertion on page 4, line 85. Exercise behavior varies widely in terms of its goals and outcomes. Patients with anorexia who are solely focused on thinness concerns often engage in exercise to lose weight or avoid weight gain. In that context, exercise behavior is not geared towards gaining muscle, but rather losing fat. Thus the phrase on line 85 conflates exercise as only muscle-related when this is, in fact, not the case.

5. I’m somewhat surprised that nowhere in the introduction is a mention of muscle dysmorphia. BDD broadly is associated with body image concerns, but when those concerns are narrowed to muscularity, individuals are often diagnosed with muscle dysmorphia. This does not need to be a main point of the introduction, but I think it’s worth a mention.

6. I find the term “male gay community” to be somewhat of a misnomer. The items on the Identification and Involvement With the Gay Community Scale actually explicitly include items mentioning gay, lesbian, and bisexual individuals, and can therefore be of varying gender identities. I would encourage the authors to remove the descriptor “male” from describing this scale. “Gay” can often be used as a blanket term for individuals that are not heterosexual, but I would also encourage the authors to refer to this construct as “sexual minority community” to fully describe its facets.

Methods

7. Continuing from my above comment, the description of the IGCS seems again to be limited to only men, but in fact, interactions with other genders within the community would also appear to count toward this scale. Please rephrase to ensure accuracy.

Results

8. Table 3 still has HOM and HEM as abbreviations.

Discussion

9. On page 32, lines 656-661, the authors argue from the place that being in a relationship means that gay men are not looking for partners. This is an untenable assumption given that polyamorous (consensual non-monogamous) relationships exist and are considered by participants to be committed. I suggest that the authors rephrase to such that men in committed relationships may be less concerned with attracting new partners.

10. The phrase “…but the internal consistencies of the translated measures merely differ from those of the originals” is confusing. Maybe rephrase to “…but the internal consistencies of the translated measures are similar to the original validation studies.”

7. PLOS authors have the option to publish the peer review history of their article (what does this mean?). If published, this will include your full peer review and any attached files.

Reviewer #3: No

---

## [Author Response · Author response to Decision Letter 1]

4 Nov 2022

Revised response to Editor and Reviewers

Manuscript: Body image disturbance and associated eating disorder and body dysmorphic disorder pathology in gay and heterosexual men: A systematic analyses of cognitive, affective, behavioral und perceptual aspects (PONE-D-21-24159)

Dear Editor and Reviewer,

thank you for the opportunity to respond to any remaining comments. We did our best to respond to your remarks. Please find detailed comments on your remarks and their implementation in the manuscript underneath each notation. The page references refer to the version of our manuscript with track changes.

Editor’s comments to the Author:

We thoroughly checked our reference list for any papers that had been retracted. Fortunately, that was not the case for any manuscript that we cited.

Reviewers' comments to the Author:

Reviewer #3: 

1. To continue the commentary on gender and sex differentiation in this paper, I appreciate the consideration that the authors have already paid, and challenges related to language differences across English and German. I would also still encourage the authors to avoid the term “male” when possible, even as an adjective, due to specific language differentiation in English. Therefore, I would encourage the removal of the sentence: “Also, we use male as the adjective of man and not to differentiate between gender and sex” and instead rephrase to avoid using male. (E.g., “…Buhlmann and colleagues found that 27% of MEN IN THE STUDY reported at least one body-related concern…” as opposed to “…Buhlmann and colleagues found that 27% of MALE PARTICIPANTS reported at least one body-related concern…”

Thank you for your valuable suggestion. We gladly accept your suggestion (“A representative German study by Buhlmann and colleagues found that 27% of men in the study reported at least one body-related concern”, p. 4, l. 67) and also deleted the term “male” throughout the rest of the manuscript (with one exception, see comment no. 6), e.g. “The BAS-2 has been validated for the use with sexual minority men and women” as opposed to “The BAS-2 has been validated for the use with a male and female sexual minority sample”, p. 16, l. 328. Accordingly, we removed the sentence: “we use male as the adjective of man and not to differentiate between gender and sex” from our manuscript.

2. “Extensive, yet narrow” is a confusing phrase that the authors use multiple times throughout the paper now. Narrow is an antonym of extensive and therefore, this is somewhat oblique in meaning. I would encourage the authors to use a different phrase. If the literature is quantitively large, but narrow in scope, that is the phrase I would use.

You are right, thanks for making us aware of that and suggesting a more specific term. We were happy to exchange the term throughout the manuscript (e.g. p. 3, l. 23).

3. On page 4, line 81-82, the authors state that no studies have compared drive for leanness between gay and heterosexual men. This is not strictly true, although their analysis do make interpretation difficult. See:

Strübel, J., & Petrie, T. A. (2019). Appearance and performance enhancing drug usage and psychological well-being in gay and heterosexual men. Psychology & Sexuality, 10(2), 132–148. https://doi.org/10.1080/19419899.2019.1574879

Thanks for making us aware of this study. We included it in our manuscript (“Concerning the drive for a lean body (i.e., a trained, tight physique), only one comparative study exists, indicating that gay men have a stronger drive for leanness than heterosexual men”, p. 6, l. 104).

4. I have a bit of a quibble with the assertion on page 4, line 85. Exercise behavior varies widely in terms of its goals and outcomes. Patients with anorexia who are solely focused on thinness concerns often engage in exercise to lose weight or avoid weight gain. In that context, exercise behavior is not geared towards gaining muscle, but rather losing fat. Thus the phrase on line 85 conflates exercise as only muscle-related when this is, in fact, not the case. 

You are right, this statement might be too restrictive. We changed the sentence to: “In contrast to the cognitive-affective component of body image, only a small number of studies have focused on the behavioral component of body image disturbance in gay men, and if so, predominantly on exercise behavior.” (p. 6, l. 108)

5. I’m somewhat surprised that nowhere in the introduction is a mention of muscle dysmorphia. BDD broadly is associated with body image concerns, but when those concerns are narrowed to muscularity, individuals are often diagnosed with muscle dysmorphia. This does not need to be a main point of the introduction, but I think it’s worth a mention.

That is a valid point. We added the following section to our introduction: “Regarding muscle dysmorphia, a subtype of BDD characterized by a pathological concern about one’s muscularity, evidence is equally limited. For instance, in a validation study of the Muscle Dysmorphic Disorder Inventory (MDDDI [39]), sexual minority men reported qualitatively higher MDDI total scores than heterosexual men [40]. Furthermore, in a recent Italian study [41] nearly 9% of sexual minority men exhibited a high risk of being diagnosed with muscle dysmorphia, which was again, higher than that found in heterosexual samples.” (p. 8, l. 154)

6. I find the term “male gay community” to be somewhat of a misnomer. The items on the Identification and Involvement With the Gay Community Scale actually explicitly include items mentioning gay, lesbian, and bisexual individuals, and can therefore be of varying gender identities. I would encourage the authors to remove the descriptor “male” from describing this scale. “Gay” can often be used as a blanket term for individuals that are not heterosexual, but I would also encourage the authors to refer to this construct as “sexual minority community” to fully describe its facets.

Thank you for bringing this up. The Identification and Involvement with the Gay Community Scale (IGCS) does mention reading gay or lesbian oriented paper or magazine, or attending gay or lesbian organizational activities (presumably as those usually address the LGBTQ+ community as a whole, and not only the gay community), but overall, the scale is explicitly “designed to measure involvement with and perceived closeness to the gay community among men who have sex with men” and not within the lesbian or any other sexual minority community (Vanable et al., 2011). This was of utterly importance to us, as the gay and the lesbian communities seem to have opposite effects on community member’s body image. While affiliation with the lesbian community seems to decrease body image concerns (e.g. Henn et al., 2019), affiliation with the gay community seems to increase body image concerns (e.g. Pachankis et al., 2020; Tylka et al., 2012). As the term “gay” can be used for all genders (as you already mentioned), we wanted to avoid ambiguousness by adding the word “male”. In doing so, we followed the APA guidelines for bias free language, that state: “The terms gay as an adjective and gay person as a noun have been used to refer to both males and females. However, these terms may be ambiguous in reference [..]. Thus, it is preferable to use gay or gay person only when prior reference has specified the gender composition of this term”. And further: “Lesbian and gay male are preferred to the word homosexual when used as an adjective referring to specific persons or groups, and the terms lesbians and gay men are preferred to homosexuals used as nouns when referring to specific persons or groups” (https://www.apa.org/pi/lgbt/resources/language). As a compromise, we deleted the term male in the term male gay community throughout the manuscript except when describing the scale to specify the gender composition of this term, as requested by the APA. Secondly, we tried to clearly differentiate between the gay and the lesbian community by describing: “Besides discrimination, it is assumed that pressure from within the gay community to be attractive and muscular might also contribute to elevated body image concerns among gay men [44,45]. This is in contrast to findings regarding the lesbian community which seems to act as a protective factor in the development of body dissatisfaction and appearance-related concerns [50,51]” (p. 9, l. 182). We sincerely hope that this compromise meets your and the APA’s request.

Methods

7. Continuing from my above comment, the description of the IGCS seems again to be limited to only men, but in fact, interactions with other genders within the community would also appear to count toward this scale. Please rephrase to ensure accuracy.

As stated above, the IGCS specifically addresses engagement with the gay community (not the lesbian or any other sexual minority community). This is even more apparent in the German translation of the scale. Possible interactions with other genders are only mentioned in one item (“How often do you attend any gay or lesbian organizational activities, such as meetings, fund-raisers, political activities, etc.?”), presumably as those events are usually not restricted to the gay community but to the LGBTQ+ community as a whole. All other items specifically mention involvement with the gay community. To emphasize the specificity of the scale, we would like to maintain our description.

8. Table 3 still has HOM and HEM as abbreviations.

Thanks for making us aware of that. We changed HOM and HEM to Gay men and Heterosexual men and apologize for the mistake.

Discussion

9. On page 32, lines 656-661, the authors argue from the place that being in a relationship means that gay men are not looking for partners. This is an untenable assumption given that polyamorous (consensual non-monogamous) relationships exist and are considered by participants to be committed. I suggest that the authors rephrase to such that men in committed relationships may be less concerned with attracting new partners.

Thank you for raising this point. We happily included this aspect in the discussion: “Lastly, it was argued that intracommunity pressure to conform to a certain body ideal stems from the wish to attract other members of the community for sexual and social relationships [44]. However, 42% of gay men in our study were in a committed relationship, which was a similar quantity as in heterosexual men. This may have lessened the pressure from the gay community on our gay sample to stay attractive since men in committed relationships may be less concerned with attracting new partners.” (p. 43, l. 830)

10. The phrase “…but the internal consistencies of the translated measures merely differ from those of the originals” is confusing. Maybe rephrase to “…but the internal consistencies of the translated measures are similar to the original validation studies.”

That is indeed a more intelligible phrase, thank you for the suggestion. We were happy to exchange the phrase in our manuscript (“We did not conduct a comprehensive validation process of the translated measures, but the but the internal consistencies of the translated measures are similar to the original validation studies.”, p. 44, l. 853).

---

## [Decision Letter · Decision Letter 2]

21 Nov 2022

Body image disturbance and associated eating disorder and body dysmorphic disorder pathology in gay and heterosexual men: A systematic analyses of cognitive, affective, behavioral und perceptual aspects

PONE-D-21-24159R2

Dear Dr. Schmidt,

We’re pleased to inform you that your manuscript has been judged scientifically suitable for publication and will be formally accepted for publication once it meets all outstanding technical requirements.

Kind regards,

Masaki Mogi

Academic Editor

PLOS ONE

Additional Editor Comments (optional):

Reviewers' comments:

Reviewer's Responses to Questions

**Comments to the Author**

1. If the authors have adequately addressed your comments raised in a previous round of review and you feel that this manuscript is now acceptable for publication, you may indicate that here to bypass the “Comments to the Author” section, enter your conflict of interest statement in the “Confidential to Editor” section, and submit your "Accept" recommendation.

Reviewer #3: All comments have been addressed

2. Is the manuscript technically sound, and do the data support the conclusions?

Reviewer #3: Yes

3. Has the statistical analysis been performed appropriately and rigorously? 

Reviewer #3: Yes

4. Have the authors made all data underlying the findings in their manuscript fully available?

Reviewer #3: Yes

5. Is the manuscript presented in an intelligible fashion and written in standard English?

Reviewer #3: Yes

6. Review Comments to the Author

Reviewer #3: (No Response)

7. PLOS authors have the option to publish the peer review history of their article (what does this mean?). If published, this will include your full peer review and any attached files.

Reviewer #3: No

---

## [Editor Report · Acceptance letter]

28 Nov 2022

PONE-D-21-24159R2 

Body image disturbance and associated eating disorder and body dysmorphic disorder pathology in gay and heterosexual men: A systematic analyses of cognitive, affective, behavioral und perceptual aspects 

Dear Dr. Schmidt:

I'm pleased to inform you that your manuscript has been deemed suitable for publication in PLOS ONE. Congratulations! Your manuscript is now with our production department. 

Kind regards, 

on behalf of

Dr. Masaki Mogi 

Academic Editor

PLOS ONE